# The CD4 transmembrane GGXXG and juxtamembrane (C/F)CV+C motifs mediate pMHCII-specific signaling independently of CD4-LCK interactions

**Mark S Lee[1], Peter J Tuohy[1], Caleb Y Kim[1], Philip P Yost[1], Katrina Lichauco[1], Heather L Parrish[1], Koenraad Van Doorslaer[1,2,3,4,5,6]\*, Michael S Kuhns[1,3,4,5,6]\***

[1]Department of Immunobiology, The University of Arizona College of Medicine, Tucson, United States; [2]School of Animal and Comparative Biomedical Sciences, The University of Arizona, Tucson, United States; [3]Cancer Biology Graduate Interdisciplinary Program and Genetics Graduate Interdisciplinary Program, The University of Arizona, Tucson, United States; [4]The BIO-5 Institute, The University of Arizona, Tucson, United States; [5]The University of Arizona Cancer Center, Tucson, United States; [6]The Arizona Center on Aging, The University of Arizona College of Medicine, Tucson, United States

**\*For correspondence:**
vandoorslaer@arizona.edu (KVD);
mkuhns@arizona.edu (MSK)

**Abstract** CD4[+] T cell activation is driven by five-module receptor complexes. The T cell receptor (TCR) is the receptor module that binds composite surfaces of peptide antigens embedded within MHCII molecules (pMHCII). It associates with three signaling modules (CD3γε, CD3δε, and CD3 ζ ζ ) to form TCR-CD3 complexes. CD4 is the coreceptor module. It reciprocally associates with TCR-CD3-pMHCII assemblies on the outside of a CD4[+] T cells and with the Src kinase, LCK, on the inside. Previously, we reported that the CD4 transmembrane GGXXG and cytoplasmic juxtamembrane (C/F)CV+C motifs found in eutherian (placental mammal) CD4 have constituent residues that evolved under purifying selection (Lee et al., 2022). Expressing mutants of these motifs together in T cell hybridomas increased CD4-LCK association but reduced CD3 ζ , ZAP70, and PLCγ1 phosphorylation levels, as well as IL-2 production, in response to agonist pMHCII. Because these mutants preferentially localized CD4-LCK pairs to non-raft membrane fractions, one explanation for our results was that they impaired proximal signaling by sequestering LCK away from TCR-CD3. An alternative hypothesis is that the mutations directly impacted signaling because the motifs normally play an LCK-independent role in signaling. The goal of this study was to discriminate between these possibilities. Using T cell hybridomas, our results indicate that: intracellular CD4-LCK interactions are not necessary for pMHCII-specific signal initiation; the GGXXG and (C/F)CV+C motifs are key determinants of CD4-mediated pMHCII-specific signal amplification; the GGXXG and (C/F)CV+C motifs exert their functions independently of direct CD4-LCK association. These data provide a mechanistic explanation for why residues within these motifs are under purifying selection in jawed vertebrates. The results are also important to consider for biomimetic engineering of synthetic receptors.

## eLife assessment

This study provides **valuable** new insights as to how two evolutionary conserved motifs in CD4 contribute to the CD4-mediated enhancement of TCR signaling independently of the CD4-LCK interaction. The data at hand are **convincing**, even if confined to a cell line model and not substantiated in vivo and with little new mechanistic insight provided regarding the domains of CD4 shown

to have significant roles in the signaling process. Without the data from primary cells it is difficult to make statements about the quantitative contribution of LCK-dependent and independent functions of CD4 in TCR signaling.

## Introduction

Delineating the mechanistic principles by which multi-module receptors drive complex biological processes is important both for our broad understanding of receptor biology and for guiding biomimetic engineering of synthetic receptors for therapeutic purposes. For example, the design of conventional single-chain chimeric antigen receptors (CARs) utilized in CAR-T cell therapy was informed by an early 1990s understanding of receptor biology and bears little resemblance to the multi-module receptors that naturally drive T cell responses. These design differences necessitate that conventional CARs integrate with the T cell intracellular signaling machinery differently than their natural counterparts, leading us to posit that fully coopting T cell functions with synthetic receptors requires that they be designed to integrate with the T cell's intracellular signaling machinery in a way that mimics the native receptors (*Harris and Kranz, 2016*; *Kobayashi et al., 2020*; *Wu et al., 2020*). Achieving this goal requires a more complete understanding of the multi-module receptors that mediate antigen-specific T cell activation. In particular, understanding how CD4$^+$ T cells respond to antigen is important for informing biomimetic engineering of CARs because recent data show that redirecting CD4$^+$ T cells with CARs is important for long-lived therapeutic efficacy (*Melenhorst et al., 2022*).

CD4$^+$ T cells are driven by five-module receptors that recognize peptide antigens embedded within MHCII (pMHCII). Each naïve CD4$^+$ T cell expresses a clonotypic receptor module, called the T cell receptor (TCR), that binds specifically to unique features of composite pMHCII surfaces (*Kuhns and Badgandi, 2012*; *Kuhns and Davis, 2012*). The TCR lacks intracellular signaling domains and instead assembles with three signaling modules (CD3γε, δε, and ζζ) that have immunoreceptor tyrosine-based activation motifs (ITAMs), and other motifs, to connect TCR-pMHCII interactions to the intracellular signaling apparatus. CD4 is the coreceptor module. It binds pMHCII in a reciprocal fashion with TCR-CD3 on the outside of CD4$^+$ T cells and interacts with the Src kinase, LCK, via an intracellular CQC zinc clasp motif and helix (*Huse et al., 1998*; *Kim et al., 2003*). According to the TCR signaling paradigm, co-engagement of pMHCII by TCR-CD3 and CD4 positions LCK and the CD3 ITAMs in the proper spatial proximity to enable LCK phosphorylation of the CD3 ITAMs to initiate signaling (*Rudd, 2021*). However, when we tested this model directly we found that CD4-LCK interactions are not key determinants of CD3ζ ITAM phosphorylation (pCD3ζ) (*Lee et al., 2022*). Based on our work and that of other labs, including a study showing that C-terminally truncated CD4 lacking the motifs that mediate CD4-LCK interactions can nevertheless drive CD4$^+$ T cell development, proliferation, and T-helper functions, we consider the question of how intracellular pMHCII-specific CD4$^+$ T cell signaling is initiated to be unresolved (*Glassman et al., 2018*; *Horkova et al., 2023*; *Killeen and Littman, 1993*; *Xu and Littman, 1993*).

The answer to this question lies with how the five modules of pMHCII receptors evolved over ~435 million years to refine pMHCII-specific signaling. We have therefore taken a systems biology approach to computationally analyze data from real-world experiments performed in 99 jawed vertebrate species under a variety of conditions over the course of evolution. By computationally reconstructing the evolutionary history of CD4, we identify residues and motifs in the extracellular, transmembrane, and intracellular domains that are of predicted to be functionally important (*Lee et al., 2022*). We then cross-validated these findings using a well-established T cell hybridoma experimental system (58α⁻β⁻ cells) wherein we performed structure-function analysis of mutants of key motifs to infer their function by evaluating the phenotypes of the mutants relative to the wild-type (WT). When we mutated the intracellular CQC clasp and IKRLL helix motifs that are known to mediate CD4-LCK interactions, we observed significant reductions in CD4-LCK association; yet we did not observe the predicted decrease in CD3ζ ITAM phosphorylation (pCD3ζ). In contrast, when we mutated the transmembrane GGXXG motif and cytoplasmic juxtamembrane (C/F)CV+C palmitoylation motif, we observed an increased frequency of CD4-LCK pairs that were preferentially localized to detergent soluble membrane (DSM) domains. We also found evidence of reduced CD3ζ, ZAP70, and PLCγ1 phosphorylation (pCD3ζ, pZAP70, and pPLCγ1) compared with WT CD4. The reason that constituent residues of these motifs evolved under purifying selection therefore appears to be due to

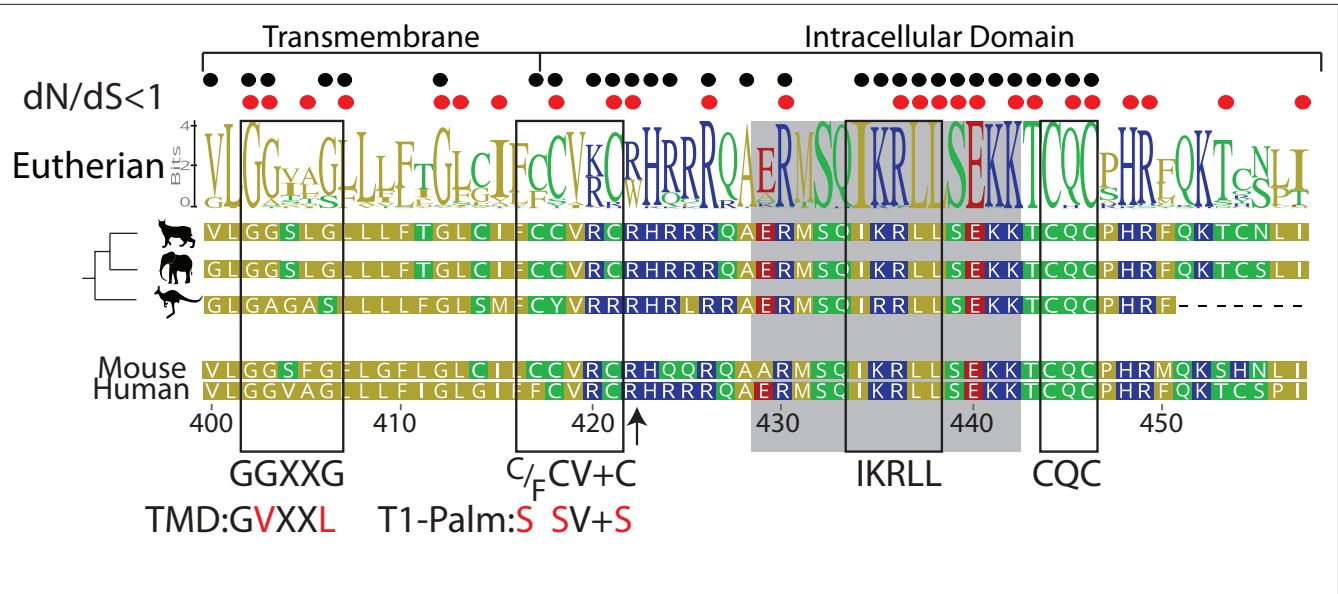

**Figure 1.** Computational reconstruction of CD4 evolution. The maximum likelihood phylogenetic tree clusters mammalian CD4 sequences. The tree highlights ancestral reconstructions of CD4 sequences from marsupials (kangaroo silhouette), Atlantogenata (elephant), and Boreoeutheria (wildcat). The logoplot of extant eutherian (Atlantogenata and Boreoeutheria) CD4 sequences show sequence conservation over evolutionary time. In these plots, the height of symbols indicates the relative frequency of each amino acid at that specific position. The mouse CD4 numbering (uniprot) is used as a reference, and residues are color-coded based on sidechain polarity. Evolutionary insertion or deletion events are indicated by dashes (-). Most recent common ancestor (MRCA) sequences are shown at each node in the tree (Nodes 1–4). As in our previous study (*Lee et al., 2022*), the ratio of synonymous (dS) and nonsynonymous (dN) substitution rates was calculated. Black dots indicate dN/dS ratios that are significantly below 1 across the entire dataset. Red dots indicate residues under purifying selection in the mammalian only dataset. Previously identified motifs are indicated by boxes, while the intracellular domain helix is shaded gray. The arrow at position 422 indicates where CD4 was truncated (CD4-T1), while TMD and T1-Palm show the mutations studied in this study.

reduced signaling in response to pMHCII when the motifs are altered. We took these data as evidence that the GGXXG and (C/F)CV+C motifs arose in eutherians (placental mammals) to tightly regulate pMHCII-specific signal initiation and postulated that the mutant CD4 molecules might sequester LCK away from TCR-CD3 to prevent CD3$\zeta$ phosphorylation. An alternative hypothesis is that the GGXXG and (C/F)CV+C motifs impact pCD3$\zeta$ levels independently of CD4-LCK interactions.

In the present study we tested these hypotheses by studying the impact of GGXXG and (C/F)CV+C motif mutants individually, or together, on proximal pMHCII-specific signaling in C-terminally truncated CD4 molecules (T1). Working with T1 eliminated motifs in the intracellular domain, including those that mediate CD4-LCK association. As expected from prior work, truncating CD4 relieved CD4-LCK interactions and reduced IL-2 production. However, there was no impact on the average pCD3$\zeta$ levels or the percent of cells with phosphorylated CD3$\zeta$ when compared with WT CD4, despite significantly reduced levels of CD4-LCK pairs. These data provide further evidence that the core tenet of the TCR signaling paradigm, wherein CD4 recruits LCK to CD3 ITAMs initiates signaling, needs revising. Importantly, T1 molecules bearing mutations in the GGXXG and (C/F)CV+C motifs individually or together reduced IL-2 production as well as pCD3$\zeta$ levels and other TCR-proximal signaling events. The simplest interpretation of these data is that the GGXXG and (C/F)CV+C motifs are key determinants of pMHCII-specific signaling on their own, independent of CD4-LCK interactions.

## Results

The goal of this study was to evaluate if the transmembrane GGXXG and juxtamembrane (C/F)CV+C palmitoylation motifs, which are highly conserved in eutherians (placental mammals) and contain constituent residues under purifying selection (*Figure 1*), influence pMHCII-specific proximal signaling on their own rather than by sequestering CD4-LCK pairs away from membrane rafts. Accordingly, we used 58$\alpha^-\beta^-$ T cell hybridomas transduced to express the 5c.c7 TCR along with either the WT or

**Table 1.** Motifs and mutants analyzed in this study.

| Motif location/known function | Mutant names | Mutated motif | Residue mutations |
|---|---|---|---|
| C-terminally truncated CD4 | T1 | Ends at R422 | R422 is the last residue |
| Extracellular D1 C"-strand (binds pMHCII) | T1$^{\Delta bind}$ | G<u>K</u>G<u>VLIR</u> | K68D, V70D, L71S, I72D, R73S |
| TMD/protein or cholesterol interactions | T1-TMD | G<u>G</u>xx<u>G</u> | G403V, G406L |
| Juxtamembrane/palmitoylation | T1-Palm (2C) | <u>C</u>V + <u>C</u> | C418S, C421S |
| Juxtamembrane/palmitoylation | T1-Palm (3C) | <u>CC</u>V + <u>C</u> | C471S,C418S, C421S |
| TMD+Palm/raft localization | T1-TP (2C) | G<u>G</u>xx<u>G</u>, <u>C</u>V+<u>C</u> | G403V, G406L, C418S, C421S |
| TMD+Palm/raft localization | T1-TP (3C) | G<u>G</u>xx<u>G</u>, <u>CC</u>V+<u>C</u> | G403V, G406L, C417S, C418S, C421S |

mutant CD4 constructs described in *Table 1*. The well-characterized 5c.c7 TCR recognizes the moth cytochrome *c* (MCC) peptide 88–103 in the context of mouse MHCII I-E$^k$. We used this experimental system because 58α$^-$β$^-$ cells were used in seminal work to link CD4-LCK association to IL-2 production (*Glaichenhaus et al., 1991*). Specifically, mutating or removing the intracellular CQC clasp motif was shown in that study to reduce both CD4-LCK association and signaling output as measured by IL-2 production. We confirmed those findings in our recent study (*Lee et al., 2022*).

## WT and T1 equivalently enhance pMHCII-specific proximal signaling

Previously we reported that the C-terminally truncated mouse CD4 T1 mutant (*Table 1* and *Figure 1* arrow) reduces but does not prevent IL-2 production relative to WT CD4 despite lacking most of the intracellular domain, including the CQC clasp and IKRLL motifs shown to mediate CD4-LCK association (*Lee et al., 2022*). Because T1 maintains both the transmembrane GGXXG and juxtamembrane (C/F)CV+C motifs, we reasoned that it had utility for studying the contributions these motifs make to pMHCII-specific signaling in the absence of CD4-LCK interactions.

Accordingly, we generated 5c.c7$^+$ 58α$^-$β$^-$ cells expressing WT CD4, T1, or T1 combined with our previously reported Δbind mutant that disrupts binding to pMHCII in the CD4 D1 domain. This T1$^{\Delta bind}$ mutant allowed us to eliminate the contributions of CD4-pMHCII interactions as well as CD4-LCK interactions (*Table 1* and *Figure 2—figure supplement 1*; *Glassman et al., 2016*; *Glassman et al., 2018*; *Parrish et al., 2015*). The goals were to: (1) determine if T1 significantly impairs pMHCII-specific signaling events, relative to WT, as expected because it lacks the CQC and IKRLL motifs that mediate CD4-LCK interactions; (2) use T1$^{\Delta bind}$ to establish the baseline above which pMHCII-dependent assembly of CD4 with TCR-CD3 contributes to signaling. The T1 mutant produced measurable IL-2 (response magnitude ~61% of WT, *Table 2*) that was consistently reduced across a range of MCC peptide concentrations compared to WT, whereas the T1$^{\Delta bind}$ mutant did not produce IL-2. We also observed reduced pMHCII-specific endocytosis for T1 and T1$^{\Delta bind}$ compared to WT (*Figure 2A* and *Figure 2—figure supplement 2*). These data indicate that, in our system, extracellular CD4-pMHCII engagement is essential for IL-2 production while the CD4 intracellular domain C-terminal of R422 contributes to IL-2 production, but is not absolutely necessary in the absence of other elements of the intracellular domain.

**Table 2.** Wildtype (WT) vs. T1 data summary.
All values presented as percent of WT control (truncation average/WT average × 100) rounded to the nearest whole number.

| | IL-2 AUC | IL-2 41 nM MCC | pCD3ζ MFI | pCD3ζ Response | pZAP70 MFI | pZAP70 Response | pPLCγ1 MFI | pPLCγ1 Response | LCK DRM AUC | LCK Total AUC | LCK DSM AUC | CD4 MFI | CD4 endocytosis |
|---|---|---|---|---|---|---|---|---|---|---|---|---|---|
| WT | 100 | 100 | 100 | 100 | 100 | 100 | 100 | 100 | 100 | 100 | 100 | 100 | 100 |
| T1 | 61 | 14 | 106 | 95 | 106 | 102 | 104 | 113 | 29 | 19 | 16 | 110 | 14 |

IL-2 area under the curve (AUC) = response magnitude; IL-2 41 nM moth cytochrome *c* (MCC) = sensitivity; pCD3 ζ , pZAP70, and pPLCγ1 MFI = intensity; pCD3 ζ , pZAP70, and pPLCγ1 response = % responders; LCK DRM (detergent resistant membrane), DSM (detergent soluble membrane), and total AUC = sucrose gradient analysis; CD4 MFI = cell surface expression intensity; CD4 endocytosis=activation-induced internalization.

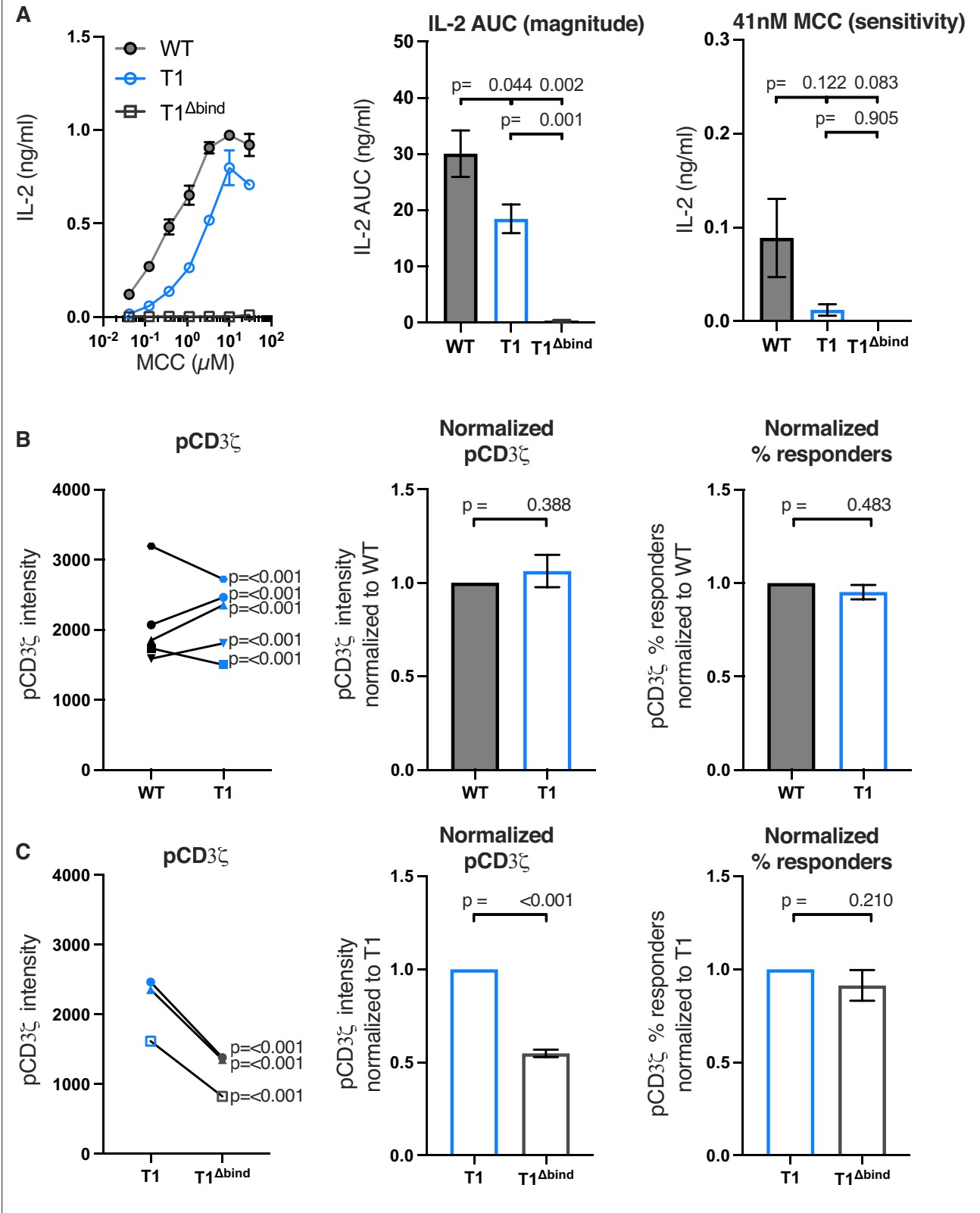

**Figure 2.** Truncating CD4 does not reduce CD3 ζ phosphorylation. (**A**) Representative IL-2 production is shown in response to a titration of moth cytochrome *c* (MCC) peptide from one experiment (left). Experiments were performed in triplicate and each symbol equals the mean± SEM at that peptide concentration. Area under the curve (AUC) analysis for the dose response is shown as a measure of the response magnitude for the average of three independent experiments performed with one independently generated set of cell lines (center). The average response to a low dose (41 nM) of

*Figure 2 continued on next page*

*Figure 2 continued*

peptide is shown as a measure of sensitivity for three independent experiments performed with one independently generated set of cell lines (right). The data are representative of those obtained with four (WT vs T1) or two (WT vs T1$^{\Delta bind}$) independently generated sets of cell lines. (**B**) Phosphorylation intensity of CD3$\zeta$ (pCD3$\zeta$) for WT and CD4-T1 (T1) (left), normalized pCD3$\zeta$ intensity for WT and T1 (center), and average % responders of pCD3$\zeta$ for WT and T1 (right). Five independently generated cell lines (biological replicates) were tested for WT and T1. For phosphorylation intensity, each pair of lines (connected symbols) was tested in three independent experiments. Data were analyzed and collected as in *Figure 2—figure supplement 3*. One-way ANOVA was performed with a Dunnett's post-test using GraphPad Prism 9. For normalized intensity, all individual mutant cell line values were normalized to their paired control values. Average % responders are presented. Bars represent the mean± SEM. One-way ANOVA with paired comparisons was performed with a Sidak's post-test for specific comparisons of normalized values using GraphPad Prism 9. All generated cells lines were considered for the multiple comparisons. (**C**) Phosphorylation intensity of pCD3$\zeta$ for T1 and T1$^{\Delta bind}$ (left), normalized pCD3$\zeta$ intensity for T1 and T1$^{\Delta bind}$ (center), and % responders of pCD3$\zeta$ for T1 and T1$^{\Delta bind}$ (right) were performed and analyzed as in B, with the exception that the open symbols represent data from a single experiment whereas the closed symbols represent aggregate data from three independent experiments. Two-tailed t-tests were performed to compare the single T1 vs T1$^{\Delta bind}$ samples as no other samples were collected in parallel.

The online version of this article includes the following figure supplement(s) for figure 2:

**Figure supplement 1.** Flow cytometry analysis of CD4 (left) and T cell receptor (TCR) (right) expression on 58$\alpha^-\beta^-$ hybridoma cells.

**Figure supplement 2.** T cell receptor (TCR) (left) and CD4 (right) endocytosis after pMHCII engagement is shown for the indicated cell lines after 16 hr coculture with antigen presenting cells (APCs) in the presence of 10 µM moth cytochrome *c* (MCC) peptide.

**Figure supplement 3.** Example of intracellular signaling analysis workflow.

**Figure supplement 4.** Raw and normalized pZAP70 and pPLCγ1analyses for WT and T1 mutant.

**Figure supplement 5.** Raw and normalized pZAP70 and pPLCγ1analyses for T1 and T1$^{\Delta bind}$ mutant.

Next, we evaluated the impact of these mutations on proximal signaling events measured by flow cytometry in response to antigen presenting cells (APCs) expressing agonist MCC:I-E$^k$ after background subtraction of signal levels in response to APCs expressing null Hb:I-E$^k$ (*Figure 2—figure supplement 3*). Our analysis of five independently generated pairs of WT and T1 cell lines showed variable difference in pCD3$\zeta$ levels. For three pairs of lines, the T1 response was higher than the WT and for two it was lower (*Figure 2B*). The averages of the T1 response normalized to the paired WT controls showed no difference between the population. There were also no differences for the average percent of T1 cells with response levels to MCC that were above background responses to the null Hb peptide compared to their paired WT controls. Similar results were observed for pZAP70 and pPLCγ1 levels (*Figure 2—figure supplement 4*). These data, summarized in *Table 2*, provide evidence that CD4-LCK interactions via the CQC clasp and IKRLL motifs are not key determinants of pMHCII-specific early signal events in this experimental system.

In comparison, the T1$^{\Delta bind}$ mutant significantly reduced pCD3$\zeta$, pZAP70, or pPLCγ1 levels compared to the T1 control but, interestingly, did not impact the percent of cells responding to

**Table 3.** T1 vs. T1 mutant data summary.
All values presented as percent of wildtype (WT) control (truncation average/WT average × 100) rounded to the nearest whole number.

| | IL-2 AUC | IL-2 41 nM MCC | pCD3ζ MFI | pCD3ζ Response | pZAP70 MFI | pZAP70 Response | pPLCγ1 MFI | pPLCγ1 Response | LCK DRM AUC | LCK Total AUC | LCK DSM AUC | CD4 MFI | CD4 endocytosis |
|---|---|---|---|---|---|---|---|---|---|---|---|---|---|
| T1 | 100 | 100 | 100 | 100 | 100 | 100 | 100 | 100 | 100 | 100 | 100 | 100 | 100 |
| T1 Δbind | 2.0 | 0.00 | 55 | 91 | 72 | 112 | 74 | 107 | ND | ND | ND | 74 | 39 |
| TMD | 54 | 17 | 77 | 94 | 88 | 99 | 88 | 107 | 62 | 93 | 111 | 131 | 4.0 |
| Palm (2C) | 58 | 117 | 76 | 94 | 86 | 93 | 92 | 98 | 274 | 139 | 131 | 80 | –98 |
| TP (2C) | 33 | 0.2 | 63 | 86 | 80 | 76 | 83 | 83 | 121 | 113 | 115 | 97 | 48 |
| Palm (3C) | 28 | 0.2 | 71 | 95 | 86 | 77 | 80 | 89 | 138 | 94 | 92 | 88 | 7.0 |
| TP (3C) | 32 | 46 | 62 | 85 | 80 | 67 | 71 | 83 | 87 | 101 | 103 | 80 | –55 |

IL-2 area under the curve (AUC) = response magnitude; IL-2 41 nM moth cytochrome *c* (MCC) = sensitivity; pCD3$\zeta$, pZAP70, and pPLCγ1 MFI = intensity; pCD3$\zeta$, pZAP70, and pPLCγ1 response = % responders; LCK DRM (detergent resistant membrane), DSM (detergent soluble membrane), and total AUC = sucrose gradient analysis; CD4 MFI = cell surface expression intensity; CD4 endocytosis=activation-induced internalization.

agonist MCC:I-E$^k$ (*Figure 2C* and *Figure 2—figure supplement 5*). Because the antibodies used to detect pCD3ζ, pZAP70, and pPLCγ1 recognize single phosphorylated tyrosines, these data indicate that CD4 enhances the number of TCR-CD3 complexes per cell with phosphorylated CD3ζ molecules at Y142 (same in human and mouse), as well as the number of phosphorylated ZAP70 and PLCγ1 molecules per cell, in a manner that is dependent on TCR-CD3 and CD4 co-engagement of pMHCII but independent of the intracellular domain C-terminal of R422. A summary of these results can be found in *Table 3*. A key takeaway is that the average pCD3ζ levels for three independently generated T1$^{\Delta bind}$ mutants was 54.86% of the T1 control (p<0.001). CD3ζ phosphorylation therefore occurs upon TCR engagement of cognate pMHCII in the absence of CD4-pMHCII interactions, and CD4 engagement of pMHCII increases the number of phosphorylated TCR-CD3 complexes. Importantly, the pCD3ζ levels observed in the T1$^{\Delta bind}$ mutants is subthreshold for IL-2 production at the bulk population level. These results allow us to conclude that pMHCII-specific baseline pCD3ζ levels achieved in our system in the absence of CD4-pMHCII interactions are approximately half of that achieved when CD4 can interact with pMHCII on the outside of the cell. Furthermore, consistent with the kinetic discrimination model of T cell activation, CD4-pMHCII interactions on the outside of the cell appear to be essential to push pCD3ζ levels above the threshold level needed for IL-2 production (*Rabinowitz et al., 1996*).

## The GGXXG and (C/F)CV+C motifs influence IL-2 production independently of LCK

Having established that T1 contributes to early pMHCII-specific signaling (e.g. pCD3ζ) as well as signaling output (i.e. IL-2), we generated 5c.c7$^+$ 58α$^-$β$^-$ cells expressing WT, T1, or T1 bearing mutants of the GGXXG and/or (C/F)CV+C motifs (*Table 1* and *Figure 3—figure supplement 1*). To study the function of the GGXXG motif alone, we used our previously described TMD mutant (GGXXG to GVXXL) wherein bulky side chains replace the glycines that compose a flat surface on the transmembrane helix that could mediate protein:protein or protein:cholesterol interactions (*Fessler, 2016*; *Parrish et al., 2015*; *Song et al., 2014*; *Teese and Langosch, 2015*; *Wacker et al., 2013*). We called these mutants T1-TMD. To test the function of the palmitoylation motif, which contains the core CVRC (418–421) sequence in mouse and humans, we made note of a broader (C/F)CVRC motif in eutherians wherein the majority of ortholog sequences had CCVRC, as found in mouse, and a minority had FCVRC as found in humans. Importantly, position 417 is under purifying selection for all extant CD4 orthologs we analyzed by the fixed effects likelihood (FEL) method (*Figure 1*), indicating that this residue evolved under purifying selection due to functional importance (*Lee et al., 2022*). We therefore made T1-Palm (3C) mutants wherein we mutated all three cysteines to serines as well as T1-Palm (2C) mutants where only the core cysteines common to mouse and humans were mutated to serines. Finally, we made mutants containing both the TMD and Palm mutants that we called T1-TP(3C) or T1-TP(2C).

To confirm that truncating CD4 relieved CD4-LCK association, and evaluate if the mutations impacted any residual CD4-LCK interactions that may occur due to colocalization in particular membrane domains, we performed sucrose gradient analysis of detergent lysates of WT, T1, T1-TMD, T1-Palm(2C), T1-Palm(3C), T1-TP(2C), and T1-TP(3C). We observed significantly reduced CD4-LCK association in detergent resistant membranes (DRMs, aka membrane rafts) and DSMs for T1 compared to WT (*Figure 3—figure supplement 2*; *Pike, 2006*). The mutants did not impact residual CD4-LCK association in the DRMs or DSMs. Furthermore, the total LCK signal associated with CD4 was significantly lower for T1 compared with the WT, and the residual association was not impacted by the mutants (*Table 3* and *Figure 3—figure supplement 3*). We therefore conclude that, in our experimental system, any functional differences between the T1-TMD, T1-Palm(2C/3C), or T1-TP(2C/3C) mutants and the T1 control is a function of the motif we are studying in the absence of direct CD4-LCK interactions.

Next, we measured IL-2 production by WT, T1, and CD4 mutant cells to evaluate the impact of the mutations on signaling output in response to MCC. Here again, T1 was slightly lower than WT, while the T1-TMD reduced the IL-2 response magnitude and sensitivity relative to the T1 control (*Figure 3A*). This was expected given our prior work showing that the TMD mutant reduced IL-2 production driven by a shorter CD4 truncation mutant (CD4T, mutated at R420) (*Parrish et al., 2015*). Mutating the GGXXG or (C/F)CV+C motifs individually or together did not impact pMHCII-induced

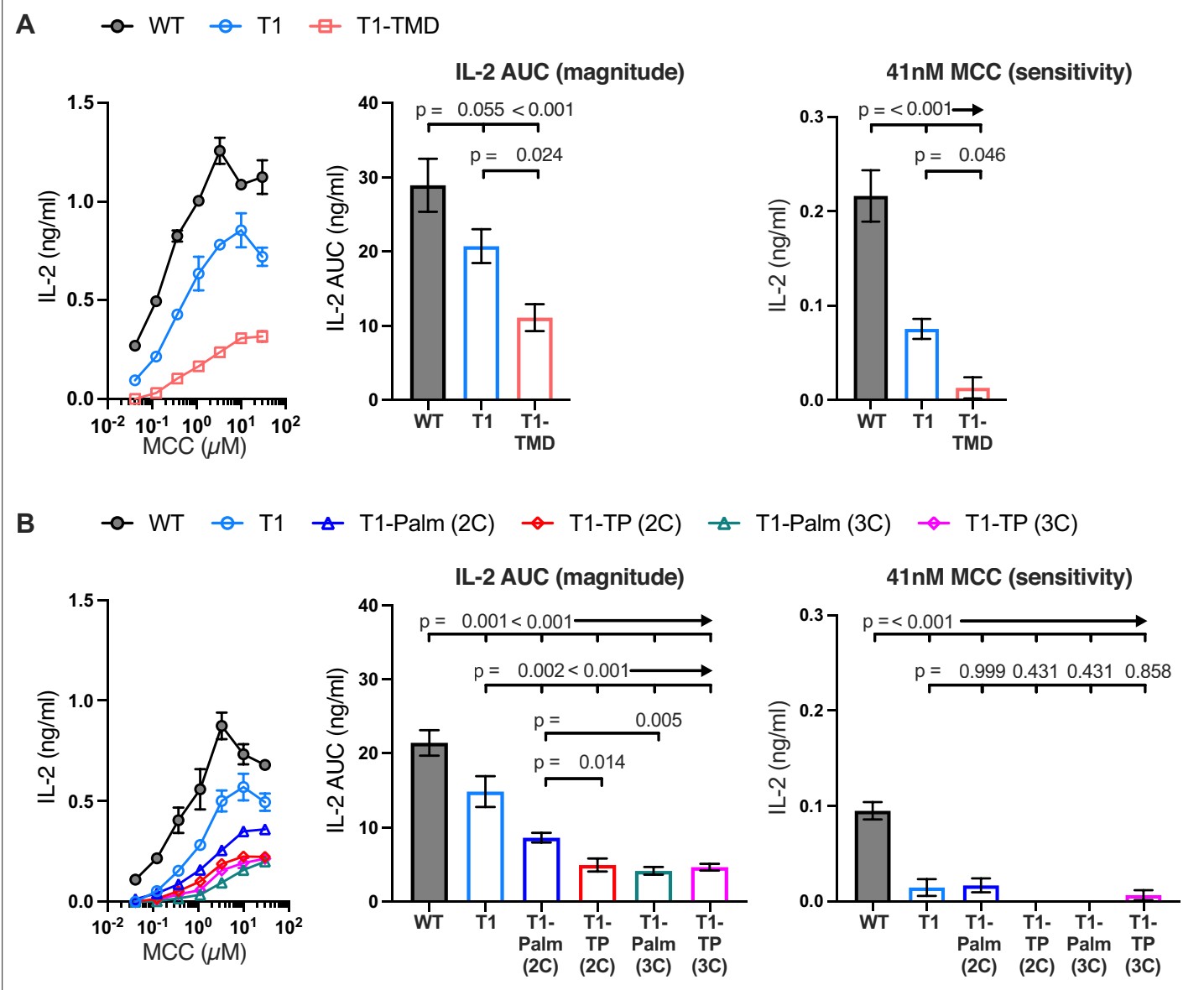

**Figure 3.** The GGXXG and CV+C motifs are key determinants of CD4 function. (**A, B**) Representative IL-2 production is shown in response to a titration of moth cytochrome *c* (MCC) peptide from one experiment (left). Experiments were performed in triplicate and each symbol equals the mean± SEM at that peptide concentration. Area under the curve (AUC) analysis for the dose response is shown as a measure of the response magnitude for the average of three independent experiments performed with one independently generated set of cell lines (center). The average response to a low dose (41 nM) of peptide is shown as a measure of sensitivity for three independent experiments performed with one independently generated set of cell lines (right). Results are representative of those obtained with at least three independently generated sets of cell lines. One-way ANOVA was performed with a Dunnett's post-test for comparisons with wildtype (WT) and T1 samples, and a Sidak's post-test for comparisons between selected samples.

The online version of this article includes the following figure supplement(s) for figure 3:

**Figure supplement 1.** Flow cytometry measurements of CD4 and TCR surface expression.

**Figure supplement 2.** Sucrose gradient analysis of CD4-Lck association.

**Figure supplement 3.** Total Lck associated with Lck (DRMs + DSMs).

**Figure supplement 4.** Activation induced TCR and CD4 endocytosis.

TCR endocytosis relative to the T1 control, and activation-induced CD4 endocytosis was inhibited for all truncated CD4 molecules (*Figure 3—figure supplement 4A and B*). The T1-Palm(2C), T1-TP(2C), T1-Palm(3C), and T1-TP(3C) all reduced IL-2 production relative to T1, indicating that they also influence the LCK-independent contribution of CD4 to pMHCII-specific signaling (*Figure 3B*). Of note, the

T1-TP(2C) IL-2 response magnitude was lower than that of T1-Palm(2C) alone, consistent with what we previously observed with the full-length versions of these CD4 mutants (*Figure 3B*; *Lee et al., 2022*). Furthermore, the T1-Palm(3C) response magnitude was lower than that of T1-Palm(2C), suggesting that the additional cysteine contributes to signaling. Together, these data indicate that the GGXXG and (C/F)CV+C motifs contribute to the signaling cascade that leads to pMHCII-specific IL-2 production when CD4 does not directly interact with LCK via the intracellular CQC clasp and IKRLL motifs. These data do not tell us where in the signaling pathway the contributions are made.

## The CD4 GGXXG and (C/F)CV+C motifs are key determinants of early signaling

To evaluate if the GGXXG and (C/F)CV+C motifs influence early signaling events, we measured pCD3$\zeta$, pZAP70, and pPLC$\gamma$1 levels for four independently generated T1-TMD lines, three independently generated T1-Palm(2C) lines, and three independently generated T1-TP(2C) lines relative to their matched T1 controls in response to APCs expressing MCC:I-E$^k$ (*Figure 4* and *Figure 4—figure supplements 1 and 2*, and *Table 3*). For the T1-TMD, T1-Palm(2C), and T1-TP(2C) mutant lines we observed lower average levels of pCD3$\zeta$, pZAP70, and pPLC$\gamma$1 relative to the T1 controls. These data indicate that the GGXXG and (C/F)CV+C motifs contribute to CD4's ability to enhance proximal signaling in the absence of CD4-LCK interactions.

Of note, when we considered the average fold change of the means of the independently generated mutant cell lines relative to their paired T1 controls, we included the mean of the fold change of the T1$^{\Delta bind}$ mutant cells from *Figure 2C* relative to their paired T1 controls for comparison (dotted line, *Figure 4B* and *Figure 4—figure supplements 1B and 2B*). Doing so provides visual boundaries within which to consider the contribution that CD4 T1, which can bind pMHCII, makes to proximal signaling in relationship to T1$^{\Delta bind}$ (dotted line) that cannot bind pMHCII. These boundaries then provide an additional guide for evaluating the impact of the mutations on proximal signaling. For example, the T1-TMD and T1-Palm(2C) showed reduced pCD3$\zeta$ levels midway between T1 and T1$^{\Delta bind}$, which equated to a reduced contribution to pCD3$\zeta$ levels made by these mutants to ~48% of the contribution to pCD3$\zeta$ levels made by T1, while the T1-TP(2C) contributed ~16% of the pCD3$\zeta$ levels contributed by T1. The double mutant therefore nearly eliminated the contribution made by T1 to pCD3$\zeta$ levels. Although these are relative comparisons, the data further suggest a role for the GGXXG and (C/F)CV+C motifs in contributing to CD4-mediated pMHCII-specific signaling in the absence of CD4-LCK interactions.

Because we had found that T1$^{\Delta bind}$ cells failed to produce IL-2 in response to high peptide concentrations in our dose-response experiments (*Figure 1A*), whereas the T1-TP(2C) mutant cells made IL-2 in response to high peptide concentrations (*Figure 3B*), we found it interesting that the early signaling responses we measured to APCs expressing high ligand densities of tethered pMHCII were similar between the T1$^{\Delta bind}$ and T1-TP(2C) cells. We therefore evaluated IL-2 production of the T1, T1$^{\Delta bind}$, and T1-TP(2C) in response to the APCs used in these signaling studies to ask if they followed the same pattern of responses to high peptide doses in our prior experiments. We found that the T1-TP(2C) cells made approximately half as much IL-2 as the T1 controls, whereas IL-2 production by the T1$^{\Delta bind}$ cells was negligible (*Figure 4—figure supplement 3*). These data indicate that the ability of CD4 to interact with pMHCII can partially overcome the loss of function of the GGXXG and (C/F)CV+C motifs at high ligand densities with respect to a downstream signaling output, such as IL-2 production, even if the early signaling events measured at 2 min after pMHCII engagement were similar.

Finally, we considered the percent of responders (i.e. those cells that signaled above background) for the mutant cell lines. We observed small decreases in responding T1-TP(2C) cells for pCD3$\zeta$ and pZAP70 relative to the control, while pPLC$\gamma$1 trended lower for this mutant. These data indicate that mutating the GGXXG and (C/F)CV+C motifs together in the absence of the intracellular domain can reduce the number of cells that initiate pMHCII-specific signaling (*Figure 4C* and *Figure 4—figure supplements 1C and 2C*). We did not observe this phenotype in our prior study when the intracellular domain was present, suggesting this is unique to the absence of the intracellular domain (*Lee et al., 2022*).

We also measured early signaling for three independently generated T1-Palm(3C) and T1-TP(3C) mutant cell lines compared to their paired T1 controls. For all lines, pCD3$\zeta$, pZAP70, and pPLC$\gamma$1 levels were significantly lower than the controls, which was reflected in the normalized fold change

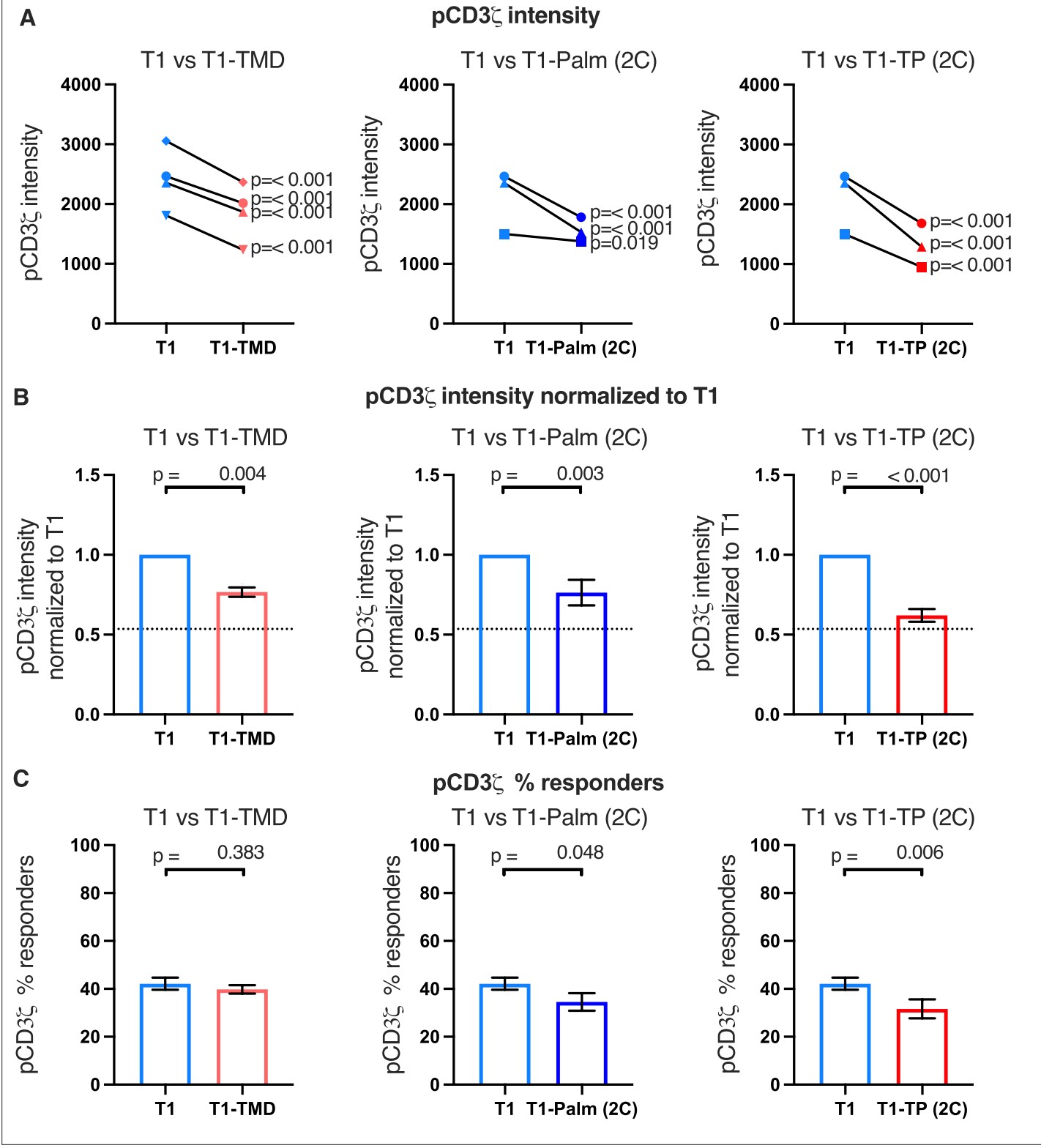

**Figure 4.** The GGXXG and CV+C motifs reduce pCD3 ζ levels. (**A**) Phosphorylation intensity of CD3 ζ for T1 and T1-TMD (left), T1 and T1-Palm (2C) (center), and T1 and T1-TP (2C) (right) are shown for independently generated pairs of (connecting line) T1 and T1-TMD (left), T1 and T1-Palm (2C) (center), and T1 and T1-TP (2C) (right) cell lines. Each pair of lines (connected closed symbols) was tested in three independent experiments (technical replicates). The data from those experiments was aggregated, and the symbols represent the mean intensity of the aggregated pCD3 ζ intensity values. One-way ANOVA was performed with a Dunnett's post-test using GraphPad Prim 9. (**B**) Data for each cell line in A are shown as the average pCD3 ζ

*Figure 4 continued on next page*

*Figure 4 continued*

intensity for all T1-TMD (left), T1-Palm (2C) (center), and T1-TP (2C) (right) cell lines normalized to their paired T1 control. Dotted line is the normalized pCD3$\zeta$ intensity for T1$^{\Delta bind}$ as a visual reference point for the contributions of CD4-pMHCII interactions. Bars represent the mean± SEM. One-way ANOVA was performed with a Sidak's post-test for specific comparisons using GraphPad Prim 9. (**C**) The average % responders for phosphorylation of CD3$\zeta$ is shown for T1-TMD (left), T1-Palm (2C) (center), and T1-TP (2C) (right) compared to the average of their paired T1 control lines. Bars represent the mean± SEM. One-way ANOVA was performed with a Sidak's post-test for specific comparisons using GraphPad Prim 9.

The online version of this article includes the following figure supplement(s) for figure 4:

**Figure supplement 1.** Raw and normalized pZAP70 analyses.

**Figure supplement 2.** Raw and normalized pPLC$\gamma$1 analyses.

**Figure supplement 3.** Representative IL-2 production is shown in response to M12 cells transduced to express tethered MCC:I-E$^k$ constructs as used in phosphorylation analysis.

(*Figure 5A and B* and *Figure 5—figure supplement 1A and B, 2A and B*). The fold change for the T1-TP(3C) levels relative to the T1 was similar to T1-TP(2C) (*Figure 5B* and *Figure 5—figure supplements 1B and 2B* and *Table 3*). These data provide further evidence that the GGXXG and (C/F)CV+C motifs together are key determinants of the LCK-independent contributions that CD4 makes to amplifying pMHCII-specific proximal signaling.

Finally, the percent responders were lower for pCD3$\zeta$ and pZAP70 for T1-TP(3C) compared to WT (*Figure 5* and *Figure 5—figure supplements 1C and 2C*). These data are consistent with the idea that, in the absence of the intracellular domain, mutating these motifs influences initiation of key signaling events. See *Table 3* for a summary of the data.

## Discussion

Our computational reconstruction of CD4 evolution provided access to a subset of results from experiments that Nature performed over ~435 million years in a greater variety of jawed vertebrates, in the wild, than could be achieved with mouse or human samples in the lab (*Lee et al., 2022*). It allowed us to identify residues that were selected under purifying selection, and are thus predicted to be functionally significant, because acquiring a mutation at these residues would affect fitness and results in a failure to propagate the change to future progeny. Our structure-function analysis of these residues in 58$\alpha^-\beta^-$ cells, which we continued in this study, provided a second system to cross-validate the functional significance of particular residues and allowed us to start providing mechanistic insights into why these residues are important.

The data presented here are not consistent with predictions of the prevailing model in which CD4 recruits LCK to CD3 ITAMs to initiate signaling. Importantly, this study builds upon our prior work by showing that while reducing CD4-LCK interactions reduces pMHCII-specific IL-2 production in our system, either by mutating the CQC clasp or removing the CD4 intracellular domain entirely, this cannot be attributed to reductions in pCD3$\zeta$ levels as predicted by the TCR signaling paradigm (*Lee et al., 2022*). It is notable that, in the absence of CD4-pMHCII interactions (T1$^{\Delta bind}$ mutant), pCD3$\zeta$ levels were roughly half of those measured when CD4-pMHCII interactions occurred (T1 mutant). We take these data together as evidence that there is a threshold for the number of phosphorylated CD3$\zeta$ molecules, and/or the duration of signaling, that is required to drive IL-2 production in our system. We also conclude that CD4 binding to pMHCII functions to increase the number of TCR-CD3 complexes per cell that experience CD3$\zeta$ phosphorylation to reach that threshold and/or sustain signaling. Furthermore, CD4-pMHCII interactions similarly impact pZAP70 and pPLC$\gamma$1 levels, albeit to a lesser extent for the latter.

A logical extension of these conclusions is that there is a density of agonist pMHCII below which CD4 binding to pMHCII is essential for signal initiation. This idea is supported by findings that CD4 is necessary for signal propagation and amplification to the calcium mobilization step in response to fewer than 25 agonist pMHCII (*Irvine et al., 2002*). Attempting to accurately measure the contributions of CD4 to ITAM phosphorylation, with or without mutant motifs, at very low densities of pMHCII would be challenging and is technically beyond the scope of this study. Nevertheless, in this and our prior publication we did not observe the expected decreases in pCD3$\zeta$ levels when CD4-LCK interactions were reduced by mutating the CQC clasp motif, the IKRLL motif, or by eliminating the intracellular domain. Furthermore, we found in our prior study that IL-2 production increased significantly

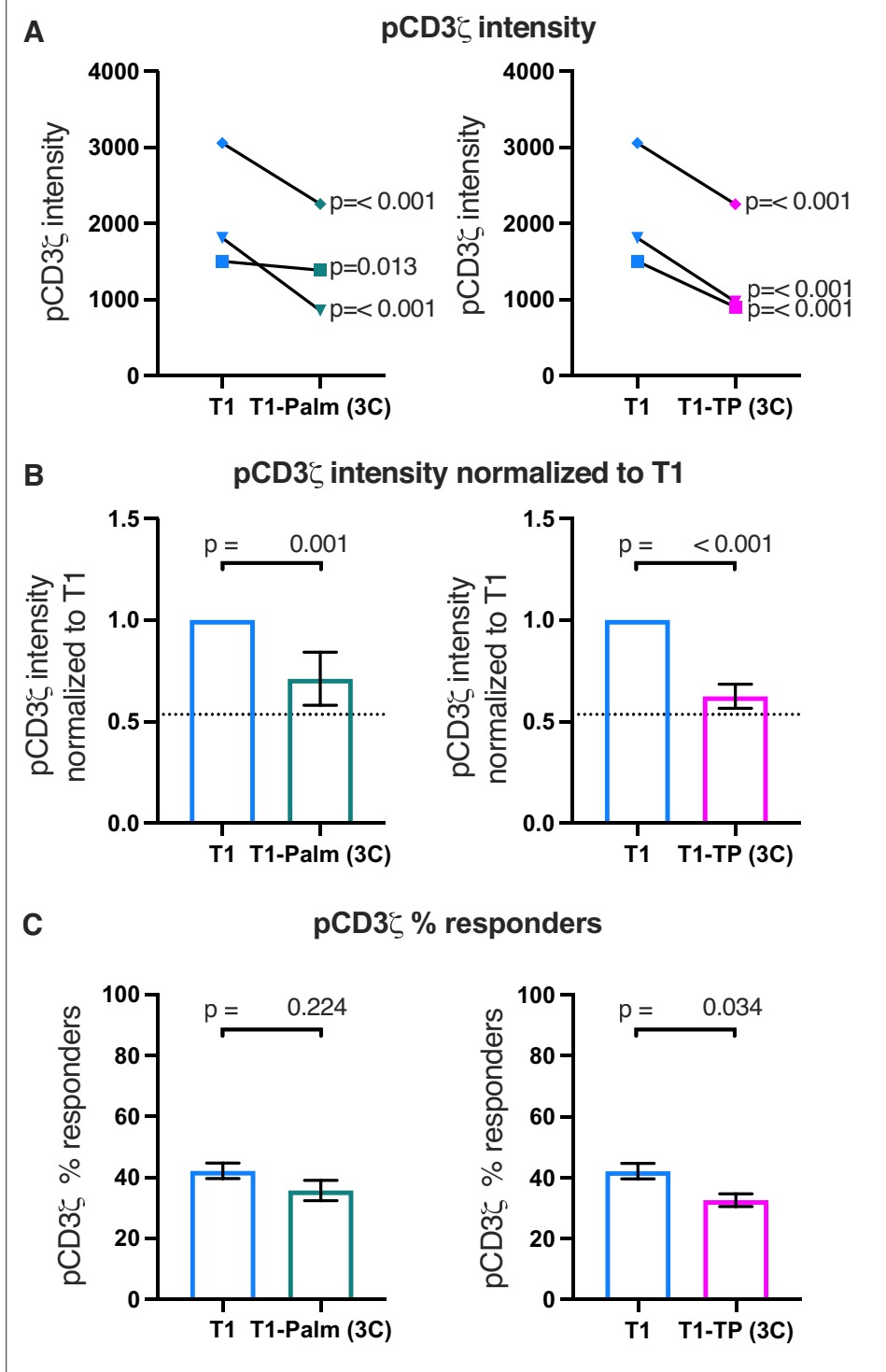

**Figure 5.** The (C/F)CVRC motifs reduce proximal T cell receptor (TCR)-CD3 signaling. (**A**) Phosphorylation intensity of CD3 $\zeta$ for T1 and T1-Palm (3C) (left) and T1 and T1-TP (3C) (right) are shown for independently generated pairs of (connecting lines) T1 and T1-Palm (3C) (left) and T1 and T1-TP (3C) (right) cell lines. Analysis was performed as in *Figure 4A*. (**B**) Data for each mutant cell line in A are shown as the average pCD3 $\zeta$ intensity values for T1-Palm (3C) (left) and T1-TP (3C) (right) normalized to their paired T1 controls. The dotted line represents the normalized phosphorylation of CD3 $\zeta$ intensity for T1$^{\Delta bind}$ as a visual reference for the contributions of CD4-pMHCII interactions. Analysis was performed as in *Figure 4C*. (**C**) Average % responders for phosphorylation of CD3 $\zeta$ is

*Figure 5 continued on next page*

*Figure 5 continued*

shown for T1-Palm (3C) (left) and T1-TP (3C) (right) are shown compared to the average of their paired T1 control. Analysis was performed as in *Figure 4C*.

The online version of this article includes the following figure supplement(s) for figure 5:

**Figure supplement 1.** Raw and normalized pZAP70 analyses.

**Figure supplement 2.** Raw and normalized analysis of pPlCγ1.

when we mutated the full intracellular helix, the helix and CQC clasp together, or the IKRLL motif alone. These direct tests of the TCR signaling paradigm are inconsistent with predictions of the model. As such, they suggest that the contributions of CD4 to signal initiation and amplification at the early timepoint we measured are independent of direct CD4-LCK interactions.

Because the combined impact of mutating the GGXXG and (C/F)CV+C motifs reduced pCD3ζ levels to ~16% of the contribution of CD4-pMHCII engagement, even when direct CD4-LCK interactions were absent, we can conclude that the GGXXG and (C/F)CV+C motifs are key determinants that help CD4 drive signaling above the threshold required for IL-2 production. Our data also provide evidence that mutating the GGXXG and (C/F)CV+C motifs together can reduce the frequency of cells experiencing CD3ζ and ZAP70 phosphorylation in the absence of the intracellular domain. Given that palmitoylation is rapidly reversible, these data suggest that the switch-like function of the palmitoylation motif may influence signal initiation. Interestingly, unlike CD4-pMHCII interactions, these motifs are not essential for signaling output at high ligand densities as we observed low levels of IL-2 produced in cells bearing mutants of these motifs with increasing doses of agonist pMHCII. Binding of CD4 and TCR-CD3 to pMHCII can therefore amplify pMHCII-specific signaling without the contributions of the GGXXG and (C/F)CV+C motifs, or motifs in the intracellular domain, albeit at lower levels than if these motifs are intact. This could be due to the contributions of additional unidentified motifs in the transmembrane domain, or to the increase in TCR-CD3 dwell time on pMHCII that is mediated by the CD4 extracellular domain (*Glassman et al., 2018*). Given that these motifs were essential for IL-2 production at low ligand densities, a logical extension of these data is that there is an agonist pMHCII density below which these motifs are essential for the initiation of early signaling events.

The data from our current study indicate that the GGXXG and (C/F)CV+C motifs do not influence early signaling events by regulating the ability of CD4 to either recruit LCK to, or sequester it away from, the TCR-CD3 ITAMs as we previously postulated (*Lee et al., 2022*). Instead, they work independently of CD4-LCK interactions to enhance pMHCII-specific signaling. It is tempting to speculate that because these motifs can regulate CD4 membrane domain localization and, because in other proteins palmitate moieties can sandwich cholesterol against the flat surface of a GG patch on a transmembrane domain helix, these motifs may allow CD4 to regulate local concentration of cholesterol or cholesterol sulfate around TCR-CD3, perhaps by taking such molecules away from TCR-CD3 to stabilize allosteric changes associated with signaling (*Chen et al., 2022*; *Fessler, 2016*; *Gil et al., 2002*; *Lee et al., 2015*; *Song et al., 2014*; *Swamy et al., 2016*; *Teese and Langosch, 2015*; *Wacker et al., 2013*; *Wang et al., 2016*). These CD4 motifs may also impact the accessibility of ITAMs for LCK phosphorylation by changing the local membrane environment around TCR-CD3 upon reciprocal engagement of pMHCII (*Aivazian and Stern, 2000*; *Xu et al., 2008*). Finally, it is worth considering that, in our prior study, we found G402 of the GGXXG motif has covaried over evolutionary time with L438 of the intracellular helix IKRLL motif that has inhibitory function, suggesting that the function of these motifs are co-evolving (*Lee et al., 2022*). In support of this idea, we reported that the signal enhancing activity of the GGXXG and (C/F)CV+C motifs together counterbalanced the inhibitory activity of the IKRLL motif with regard to IL-2 production. It is unclear at this point if this counterbalancing is simply additive of individual motif functions, or if there is a functional interplay of the motifs when the intracellular domain of CD4 is present. Delineating the mechanistic details by which the GGXXG and (C/F)CV+C motifs increase signaling on their own, or when combined, and how their function integrates with those of other motifs will be the subject of future investigations.

Altogether, the data in this and our preceding study provide direct evidence that complement indirect results indicating that pMHCII-specific signaling is not initiated, or dependent on, CD4 recruitment of LCK kinase activity to the CD3 ITAMs (*Glassman et al., 2018*; *Horkova et al., 2023*; *Killeen and Littman, 1993*; *Lee et al., 2022*; *Xu and Littman, 1993*). Recent work on CD8 suggest that this

coreceptor also does not recruit LCK to phosphorylate CD3 ITAMs (*Casas et al., 2014*; *Wei et al., 2020*). In total, these findings suggest that the TCR signaling paradigm needs revising as it pertains to coreceptor-LCK interactions. For CD4, the existing evidence suggest that reciprocal binding of CD4 and TCR-CD3 enables free LCK and FYN to initiate signaling by phosphorylating the ITAMs and ZAP70 (*Glassman et al., 2016*; *Glassman et al., 2018*; *Horkova et al., 2023*; *Lee et al., 2022*; *Salmond et al., 2009*; *van Oers et al., 1996*). The contribution of the current study is in showing the GGXXG and (C/F)CV+C motifs work together to enhance these early signaling events independently of direct CD4-LCK interactions.

Finally, it is interesting to consider what the primary purpose of CD4-LCK interactions via the CQC clasp and IKRLL motifs are if not to recruit LCK kinase activity to the CD3 ITAMs. Data from two studies suggest that LCK makes a kinase-independent contribution to pMHCII-specific responses; it is therefore plausible that the phenotype reported for CD4 when the CQC clasp is mutated is due to relieving the kinase-independent scaffolding function of LCK (*Horkova et al., 2023*; *Xu and Littman, 1993*). In addition, CD4-LCK interaction via the CQC clasp motif and the intracellular helix will prevent the helix from interacting with other partners, thus mutating the CQC clasp would favor helix interactions with other putative partners (*Lee et al., 2022*). Finally, the CQC clasp is thought to play a role in positioning CD4 in proximity to LAT, either through direct or indirect interactions, which could contribute to the phenotype of CQC clasp mutants (*Bosselut et al., 1999*; *Lo et al., 2018*; *Lo et al., 2019*). If CD4-LCK and CD4-LAT interactions are mutually exclusive, and both require the CQC clasp, then mutating this motif would have a greater impact than just relieving CD4-LCK interactions (*Bosselut et al., 1999*). These questions remain outstanding, as do questions concerning how the intracellular helix regulates pMHCII-specific signaling.

In closing, the broader implications of the data presented in our current and prior evolution-structure-function studies provide fertile ground for future directions beyond those mentioned above. Our approach allows us to identify functionally important residues/motif and then interrogate basic mechanistic principles of their function that should hold true across CD4+ T cell subsets since the biochemical and biophysical bases for interactions between CD4 and its interacting partners (i.e. other proteins or membrane components) should be the same in 58α−β− cells as in thymocytes or in different CD4+ T cells. However, different CD4+ T cell populations will have different levels of expression of relevant proteins (e.g. interacting partners or modifying enzymes), and may even have differences in lipid composition, such that the impact of the motifs we have identified and studied to date may lead to different outcomes in different CD4+ T cell subsets (http://immpres.co.uk/) (*Tuosto and Xu, 2018*). For example, differences in expression of enzymes with switch-like functions, such as those that add or remove palmitate to the (C/F)CV+C motif, or those that phosphorylate or dephosphorylate the serines in the intracellular helix that regulate CD4-LCK interactions and inhibitory activity, may vary between naïve CD4+ T cells, different Th subsets, or Tregs to differentially tune the activity of these motifs and their impact on pMHCII-specific signaling. Now that we have identified residues and broader motifs that have proven to be functionally important in vivo over 435 million years of evolution, while also providing mechanistic insights as to why the residues are important, there is utility in evaluating the impact of mutant CD4 mice on thymocyte development, naïve CD4+ T cell activation and differentiation, and the execution of Th and Treg effector functions. In so doing, we will gain additional insights into why there is a fitness cost if a mutation is acquired in these residues. Such knowledge will increase our fundamental understanding of CD4+ T cell biology and will also be critical for biomimetic engineering of synthetic receptors that can redirect CD4+ T cell activity for therapeutic purposes.

## Materials and methods
### Evolutionary analyses
Available CD4 orthologs were identified through reciprocal blast-based searches and downloaded from GenBank. BLAST may not only identify orthologs, so additional criteria were used to include putative orthologous CD4 sequences in our analyses: presence of a domain structure consisting of four extracellular Ig domains followed by a transmembrane domain and a C-terminal intracellular domain, including the presence of the LCK binding clasp (CxC). Sequences that were shorter, contained frameshift mutations, or displayed high sequence variability were excluded from the analysis. For the

current study, teleost fish were considered to be the oldest living species that contain a CD4 molecule given that a CD4 ortholog was not reported in the elephant shark (*Callorhinchus milii*), although future analyses of other cartilaginous fishes might yield more distant orthologs (*Venkatesh et al., 2014*). The final dataset contained 99 unique CD4 orthologs, ranging from teleost fish to human. These (putative) coding sequences were translated to amino acids and aligned using MAFFT (*Katoh et al., 2002*). For codon-based analyses, the aligned amino acid sequences were back-translated to nucleotides to maintain codons. The multiple sequence alignments were further processed to remove all insertions (indels) relative to the mouse CD4 sequence (NM_013488.3) to maintain consistent numbering of sites. The 5' and 3' regions of the CD4 molecules were not consistently aligned due to different start codon usage or extensions of the ICD, respectively. The alignment was edited to start at the codon (AAG) coding for K48 within mouse CD4. The alignment that includes all 99 CD4 sequences ends at the last cysteine residue that makes up the CQC clasp. For the mammalian only dataset, the alignment terminates at the mouse CD4 stop codon.

FastTree was used to estimate maximum likelihood trees (*Price et al., 2010*). For amino acid-based trees, the Jones-Taylor-Thornton (JTT) model of evolution was selected, while the general time-reversible model was used for nucleotide trees (*Jones et al., 1992*; *Waddell and Steel, 1997*). In all conditions, a discrete gamma model with 20 rate categories was used. A reduced representation tree (based on the amino acid alignment) is shown in *Figure 1A*. We used the same JTT model to estimate the marginal reconstructions of nodes indicated in *Figure 1A*. Phylogenetic trees and logo plots were visualized in Geneious and further edited using Adobe Illustrator. Ancestral sequences were estimated using GRASP (*Foley et al., 2020*).

Codon-based analysis of selection was performed using the hypothesis testing FEL model as implemented within the phylogenies (HyPhy) package (version 2.5.14) (MP) (*Kosakovsky Pond et al., 2020*; *Pond et al., 2005*; *Kosakovsky Pond and Frost, 2005*; *Weaver et al., 2018*). The back-translated codon-based alignments described above were used for these analyses. FEL uses likelihood ratio tests to assess a better fit of codons that allowed selection (p<0.1). When calculating values for all CD4 orthologs included in the initial phylogenetic analysis, we analyzed and identified the sequences within the mammalian clade as the foreground branches on which to test for evolutionary selection in order to maximize statistical power.

Covariation between protein residues was calculated using the MISTIC2 server. We calculated four different covariation methods (MIp, mFDCA, plmDCA, and gaussianDCA) (*Colell et al., 2018*). Protein conservation scores were calculated based on the protein alignment using the ConSurf Server (*Ashkenazy et al., 2016*; *Ashkenazy et al., 2010*). ConSurf conservation scores are normalized, so that the average score for all residues is 0, with a standard deviation of 1. The lower the score, the more conserved the protein position. For the purpose of this study, residues were considered to covary if the MI was larger than 4 and both residues had a ConSurf conservation score lower than –0.5. Also, pairs with an MI larger than 8 were considered to covary if the conservation score was below –0.3. Using these criteria, we selected 0.5% of all possible pairs as recommended (*Buslje et al., 2009*; *Colell et al., 2018*).

Statistical analyses and associated raw data for the figures are available on Dryad (doi:10.5061/dryad.p8cz8w9vw).

## Cell lines

$58\alpha^-\beta^-$ T cell hybridoma lines were generated from Kuhns Lab stocks of parental $58\alpha^-\beta^-$ T cell hybridoma cells (obtained from YH Chien at Stanford University) by retroviral transduction and maintained in culture by standard techniques as previously described (*Glassman et al., 2018*; *Letourneur and Malissen, 1989*). $58\alpha^-\beta^-$ T cell hybridomas lack expression of endogenous TCRα and TCRβ chains, are CD4 negative, make IL-2 in response to TCR signaling, and are variant of the DO-11.10.7 mouse T cell hybridoma (Balb/c T cell fused to BW5147 thymoma) (*Letourneur and Malissen, 1989*). We validate the cells lines by these characteristics as well as expression of $H2-D^d$ to validate Balb/c origin. In brief, 1 day after transduction the cells were cultured in 5 µg/mL puromycin (Invivogen) and 5 µg/mL zeocin (Thermo Fisher Scientific) in RPMI 1640 (Gibco) supplemented with 5% FBS (Atlanta Biologicals or Omega Scientific), penicillin-streptomycin-glutamine (Cytiva), 10 µg/mL ciprofloxacin (Sigma), and 50 µM beta-2-mercaptoethanol (Thermo Fisher Scientific). The next day drug concentrations were increased to 10 µg/mL puromycin (Invivogen) and 100 µg/mL zeocin (Thermo Fisher Scientific) in

10 mL in a T25 flask. Aliquot of $1\times10^7$ cells were frozen at days 5, 7, and 9. Cells were thawed from the day 5 freeze and cultured for 3 days in 10 µg/mL puromycin and 100 µg/mL zeocin, and maintained below $1\times10^6$ cells/mL to use in the functional assays. Cells used in the functional assays were grown to $0.8\times10^6$ cells/mL density and replicates of three functional assays were performed every other day. If cells exceeded $1\times10^6$ cells/mL at any point in the process they were discarded as they lose reactivity at high cell densities and a new set of vials was thawed. Typically, two independent WT and mutant pairs were generated for any given mutant and tested for IL-2 to gain further confidence in a response phenotype. When cells lines are presented together in a graph, that indicates that the cell lines (WT and mutants sets) were generated at the same time from the same parental cell stock.

Given the number of mutant CD4 cell lines generated and handled in this study, the identity of the transduced CD4 gene was verified by PCR sequencing at the conclusion of three independent functional assays. $58\alpha^-\beta^-$ T cell hybridomas were lysed using DirectPCR Tail Lysis Buffer (Viagen Biotech) with proteinase K (SIGMA) for 2 hr at 65°C. Cells were then heated at 95°C for 10 min. Cell debris was pelleted, and the supernatants were saved. CD4 was amplified by PCR using Q5 DNA Hot Start Polymerase (New England BioLabs) in 0.2 µM dNTP, 0.2 µM primer concentration, Q5 Reaction buffer, and water. CD4 was amplified using the following primers:

5' primer: acggaattccgctcgagcgccaccatggtgcgagccatctctctcttagg.

3' primer: ctagcaagcttgtcgactcaagatcttcattagatgagattatggctcttctgc.

Product were purified using SpinSmart Nucleic Acid Purification Columns (Thomas Scientific) and sent to Eton Bioscience for sequencing with the following 5' CD4 primer: gtctctgaggagcagaag.

The I-E$^{k+}$ M12 cells used as APCs were previously reported (*Glassman et al., 2018*). M12 cells are a murine B cell lymphoma from Balb/c mice (H2-D$^d$ validated) (*Kim et al., 1979*). Cells were cultured in RPMI 1640 (Gibco) supplemented with 5% FBS (Atlanta Biologicals or Omega Scientific), penicillin-streptomycin-glutamine (Cytiva), 10 µg/mL ciprofloxacin (Sigma), 50 µM beta-2-mercaptoethanol (Thermo Fisher Scientific), and 5 µg/mL puromycin (Invivogen), and 50 µg/mL zeocin (Thermo Fisher). The parental cells are maintained in Kuhns Lab stocks and were originally obtained from MM Davis stocks (Stanford University).

Parental $58\alpha^-\beta^-$ T cell hybridoma and M12 cells are periodically treated with Plasmocin and tested for mycoplasma contamination by PCR using primer sequences available from ATCC (5' primer sequence: TGCACCATCTGTCACTCTGTTAACCTC; 3' primer sequence: GGGAGCAAACAGGATT AGATACCCT). All transduced cell lines used here were grown in ciprofloxacin and generated from parental cells that had tested negative for mycoplasma.

For retroviral production we used Phoenix-eco cells from the Nolan Lab (ATCC CRL-3214).

## Antibodies

| Antibodies | Vendor | Catalog number | RRID |
|---|---|---|---|
| Anti-mouse CD4 eFlour 450, clone GK1.5 | Thermo Fisher Scientific | 48-0041-82 | RRID:AB_10718983 |
| Anti-mouse TCRα APC, clone RR8-1 | Thermo Fisher Scientific | 17-5800-82 | RRID:AB_19853170 |
| Anti-mouse CD3ε PE-Cy7, clone 145-2C11 | Thermo Fisher Scientific | 25-0031-82 | RRID:AB_469572 |
| Anti-mouse IL-2, clone JES6-1A12 | BioLegend | 503702 | RRID:AB_315292 |
| Biotin anti-mouse IL-2, clone JES6-5H4 | BioLegend | 503804 | RRID:AB_315298 |
| Streptavidin HRP | BioLegend | 405210 | |
| Anti-mouse TCRβ PE, clone KJ25 | BD Pharmingen | 553209 | RRID:AB_394709 |
| Biotin anti-mouse CD4 (Clone RM4-4) | BioLegend | 116010 | RRID:AB_2561504 |
| Anti-mouse CD4 APC, clone GK1.5 | BioLegend | 100412 | RRID:AB_312697 |
| Anti-mouse LCK PE, clone 3A5 | Santa Cruz | sc-433 | |
| Cholera Toxin Subunit B Alexa Fluor 488 | Thermo Fisher Scientific | C22841 | |
| Anti-mouse pCD3 ζ Alexa Flour 647, clone K25-407.69 | BD Phosflow | 558489 | RRID:AB_647152 |
| Anti-mouse pZAP70 APC, clone n3kobu5 | Thermo Fisher Scientific | 17-9006-42 | RRID:AB_2573268 |
| Anti-mouse pPLCγ1 PE, clone A17025A | BioLegend | 612404 | RRID:AB_2801120 |

## Flow cytometry

Cell surface expression of CD4 and TCR-CD3 complexes were measured by flow cytometry. In brief, cells were stained for 30 min at 4°C in FACS buffer (PBS, 2% FBS, and 0.02% sodium azide) using anti-CD4 (clone GK1.5, eFluor 450 conjugate, Thermo Fisher Scientific), anti-TCRα (anti-Vα11, clone RR8-1, APC conjugate, Thermo Fisher Scientific), anti-CD3ε (145-2C11, Thermo Fisher Scientific) and GFP was detected as a measure of the TCRβ-GFP subunit. Analysis was performed on a Canto II or LSRII (BD Biosciences) at the Flow Cytometry Shared Resource at the University of Arizona. Flow cytometry data were analyzed with FlowJo Version 9 software (Becton, Dickinson & Company).

## Functional assays

IL-2 production was measured to quantify pMHCII responses. $5×10^4$ transduced 58α⁻β⁻ T cell hybridomas were cocultured with $1×10^5$ transduced I-E$^{k+}$ M12 cells in triplicate in a 96-well round-bottom plate in RPMI with 5% FBS (Omega Scientific), Pen-Strep+L-Glutamine (Cytiva), 10 ng/mL ciprofloxacin (Sigma), and 50 μM beta-2-mercaptoethanol (Fisher) in the presence of titrating amounts of MCC 88-103 peptide (purchased from 21st Century Biochemicals at >95% purity) starting at 30 μM MCC and a 1:3 titration (*Glassman et al., 2018*). For experiments with APCs expressing tethered pMHCII, $5×10^4$ 58α⁻β⁻ T cell hybridomas were cultured with $1×10^5$ MCC:I-E$^{k+}$ or T102S:I-E$^{k+}$ M12 cells in triplicate in a 96-well round-bottom plate using the same culture conditions as above. The supernatants were collected and assayed for IL-2 concentration by ELISA after 16 hr of coculture at 37°C. Anti-mouse IL-2 (clone JES6-1A12, BioLegend) antibody was used to capture IL-2 from the supernatants, and biotin anti-mouse IL-2 (clone JES6-5H4, BioLegend) antibody was used as the secondary antibody. Streptavidin-HRP (BioLegend) and TMB substrate (BioLegend) were also used.

To assess engagement-induced endocytosis, CD4 surface levels were measured by flow cytometry 16 hr after coculture with APCs and peptide as described above for IL-2 quantification. 96-well plates containing cells were washed with ice-cold FACs buffer (PBS, 2% FBS, 0.02% sodium azide), transferred to ice, and Fc receptors were blocked with Fc block mAb clone 2.4G2 for 15 min at 4°C prior to surface staining for 30 min at 4°C with anti-CD4 (clone GK1.5 EF450, Invitrogen) and anti-Vβ3 TCR clone (clone KJ25, BD Pharmingen) antibodies. Cells were washed with FACS buffer prior to analysis on a LSRII (BD Biosciences) at the Flow Cytometry Shared Resource at the University of Arizona. Flow cytometry data were analyzed with FlowJo Version 10 software (Becton, Dickinson & Company). The average of the geometric mean of the TCR or CD4 signal was taken for the triplicate of the post 58α⁻β⁻ cells cocultured with M12 I-E$^{k+}$ cells at 0 μM MCC concentration. Each value of the raw gMFI of TCR or CD4 for cells cultured at 10 μM MCC was subtracted from the average gMFI at 0 μM. The values show the change of gMFI from 0 μM to 10 μM.

## Sucrose gradient analysis

Membrane fractionation by sucrose gradient was performed similarly to previously described methods (*Hur et al., 2003*; *Parrish et al., 2016*). For cell lysis, $6×10^7$ 58α⁻β⁻ T cell hybridomas were harvested and washed 2× using TNE buffer (25 mM Tris, 150 mM NaCl, 5 mM EDTA). Cells were lysed on ice in 1% Triton X detergent in TNE in a total volume of 1 mL for 10 min and then dounce homogenized 10×. The homogenized lysate was transferred to 14×95 mm² Ultraclear Ultra Centrifuge tubes (Beckman). The dounce homogenizer was rinsed with 1.6 mL of the 1% lysis buffer, which was then added to the ultracentrifuge tube. 2.5 mL of 80% sucrose was added to the centrifuge tube with lysate and mixed well. Gently, 5 mL of 30% sucrose was added to the centrifuge tubes, creating a 30% sucrose layer above the ~40% sucrose/lysate mixture. Then, 3 mL of 5% sucrose was added gently to the centrifuge tube, creating another layer. The centrifuge tubes were spun 18 hr at 4°C in a SW40Ti rotor at 36,000 rpm.

Analysis of membrane fractions was performed via flow-based fluorophore-linked immunosorbent assay (FFLISA) as previously described (*Parrish et al., 2016*). In brief, 88 μL of Streptavidin Microspheres 6.0 μm (Polysciences) were coated overnight at 4°C with 8 μg of biotinylated anti-CD4 antibody (clone RM4-4, BioLegend). Prior to immunoprecipitation (IP), beads were washed with 10 mL of FACS buffer (1× PBS, 2% FBS, 0.02% sodium azide) and resuspended in 3.5 mL of 0.1% Triton X-100 lysis wash buffer in TNE. For each cell line lysed, 10 FACS tubes were prepared with 50 μL of the washed RM4-4-coated beads. Upon completion of the spin, 500 μL was carefully taken off the top of the centrifuge tubes and discarded. Following this, 1 mL was extracted from

the top of the tube, carefully as to not disrupt the gradient, and added to a FACS tube with coated beads and capped. This was repeated for 10 individual fractions in separate FACS tubes. Following the extraction, lysates were incubated with the beads for 90 min, inverting the tubes to mix every 15 min.

Following the IP, FACS tubes were washed 3× using 0.1% Triton X lysis wash buffer in TNE. Tubes were then stained using 1 µL anti-CD4 (APC conjugate; clone GK1.5, BioLegend), 1.5 µL anti-LCK (PE conjugate, clone 3A5, Santa Cruz Biotechnology), and 1 µL CTxB (AF 488 conjugate, Thermo Fisher Scientific, resuspended as per the manufacturer's instructions) for 45 min at 4°C. Following the stain, tubes were washed using 0.1% Triton X lysis wash buffer in TNE. Analysis of beads was performed on a LSRII (BD Biosciences) at the Flow Cytometry Shared Resource at the University of Arizona. $10^4$ events were collected per sample. Flow cytometry data were analyzed with FlowJo Version 10 software (Becton, Dickinson & Company).

For FFLISA analysis, raw gMFI values for fraction 1 were subtracted from the rest of the fractions to account for background, such that the gMFI of fraction 1 is 0. To normalize the data, the percentage of CD4 within any given fraction (fx) relative to the total CD4 gMFI (CD4 signal % of total) was calculated by dividing the gMFI signal in a given fraction (fx) by the sum of the total CD4 gMFI signal [sum(f1:f10)CD4 gMFI] and multiplying by 100 [e.g. fx % of total = fx CD4 gMFI/ Sum(f1:f10)CD4 gMFI × 100]. To normalize the CTxB and LCK signal in any given fraction relative to the CD4 signal in that same fraction (CTxB or LCK normalized to CD4) the gMFI of CTxB or LCK in fx was divided by the CD4 gMFI of that fx and then multiplied by the percentage of CD4 within fx (e.g. Normalized fx LCK = fx LCK gMFI/fx CD4 gMFI × fx CD4% of total CD4 gMFI). Area under the curve (AUC) analysis was performed with GraphPad Prism 9 for fractions 1–6 to determine the AUC for the DRM domains due to their floating phenotypes, and for fractions 6–10 to determine the AUC for the DSM domains.

## Intracellular signaling analysis

M12 cells expressing Hb:I-E$^k$ (null) or MCC:I-E$^k$ (cognate) tethered pMHCII complexes were labeled with Tag-it Violet according to the manufacturer's instructions (BioLegend). M12 cells and 58α$^-$β$^-$ cells were then chilled on ice for 30 min, 5×10$^5$ of each cell type were mixed together in 1.5 mL snap cap tubes, and the cells were pelleted at 2000 rpm for 30 s at 4°C to force interactions. The supernatant was removed and the tubes were transferred to a 37°C water bath for 2 min to enable signaling. Fixation Buffer (BioLegend Inc) was then added for 15 min at 37°C. Cells were washed twice with FACs buffer, pelleted at 350×$g$ for 5 min at room temperature, resuspended in 1 mL True-Phos Perm Buffer (BioLegend Inc), and incubated at –20°C for 16 hr.

Cells were blocked with anti-mouse FcRII mAb clone 2.4G2 hybridoma supernatants (ATCC) for 30 min, pelleted, and stained on ice for 60 min with anti-pCD3 ζ (clone K25-407.69, Alexa Fluor 647 conjugate, BD Biosciences) in one sample tube, or with anti-pZAP70 (clone n3kobu5, APC conjugate, Invitrogen) and anti-pPLCγ1 (clone A17025A, PE conjugate, BioLegend) in a separate sample tube at the vendor-recommended concentrations. Finally, cells were washed 2× with FACs buffer at 1000×$g$ for 5 min at room temperature and analyzed on a Canto II (BD Biosciences) at the Flow Cytometry Shared Resource at the University of Arizona or on a BD Fortessa. 1×10$^4$ 58α$^-$β$^-$ cell:M12 cell couples were collected per sample.

Flow cytometry data were analyzed with FlowJo Verison 10 software (Becton, Dickinson, & Company) by gating on 58α$^-$β$^-$ and M12 cell couples, as described previously (*Glassman et al., 2018*). Histograms of the pCD3 ζ , pZAP70, or pPLCγ1 intensity for the gated population were then generated and data expressing the gated populations as numbers of cells within intensity bins was exported from FlowJo into Microsoft Excel where the number of cells for each bin intensity value for MCC:I-E$^k$ stimulated cells was subtracted from Hb:I-E$^k$ stimulated cells on a bin-by-bin basis. This allowed us to enumerate the intensity differences per bin upon stimulation with the agonist pMHCII over background. Mean intensity and standard error of the mean (SEM) were calculated based on the background-subtracted (MCC:I-E$^k$-Hb:I-E$^k$) data. The data was then transferred to Prism 9 where we performed smoothing analysis with 500 nearest neighbors to smooth the line profile for graphing purposes. Those intensity bins with positive values were considered to contain cells that had responded to the MCC:I-E$^k$ stimuli above background.

## Statistical analysis

Statistical analyses of sucrose gradient and functional assays were performed with GraphPad Prism 9 software as indicated in the figure legends. For each functional assay (IL-2 production and CD4 endocytosis), each individual experiment (biological replicate) was performed with triplicate analysis (technical replicates) and each experiment was repeated at least three times (three biological replicates) or as indicated in figure legends. For sucrose gradient analysis, $10^4$ beads were collected by flow cytometry in each experiment (technical replicates) and each experiment was performed three times (biological replicates). Three biological replicates were chosen for each analysis as per convention, and no power calculations were determined. One-way ANOVA was performed with a Dunnett's post-test when all mutants tested in an experiment were compared to a control sample (e.g. WT). Sidak's post-test were applied when comparing between two specific samples. These post-tests were chosen based on Prism recommendations. Student's unpaired t-tests (two-tailed) were performed when comparing WT and LL mutant samples only for phosphorylation analysis.

## Constructs

The sequences of 5cc7α, 5cc7βG, CD4 WT, and CD4 T1 constructs were previously described (*Lee et al., 2022*). Full-length CD3δ, -ε, -γ, and - ζ were encoded on a poly-cistronic construct as previously described (*Holst et al., 2008*; *Kuhns and Davis, 2007*).

The following CD4-T1 constructs: T1-TMD, T1-Palm (2C), T1-Palm (3C), T1-TP (2C), and T1-TP(3C) were cloned by conventional molecular biology techniques, including PCR-based mutagenesis where needed, into pUC18 via 5' EcoRI and 3' HindIII. After sequence verification they were subcloned into pP2 puromycin resistance MSCV vector (MCS-IRES-puro) via 5' XhoI and 3' BglII, midi-prepped (QIAGEN), and sequence-verified. Mutated codons are shown in bold uppercase letters, while the motif under interrogation is shown in bold and underlined (any non-mutated codons within the motif are bold lowercase letters and underlined). The sequences for these constructs are as follows:

## T1-TMD

acggaattccgctcgagcgccaccatggtgcgagccatctctcttaggcgcttgctgctgctgctgcagctgtcacaactcctag ctgtcactcaagggaagacgctggtgctggggaaggaagggggaatcagcagaactgccctgcgagagttcccagaagaagatca cagtcttcacctggaagttctctgaccagaggaagattctggggcagcatggcaaaggtgtattaattagaggaggttcgccttcgca gtttgatcgtttttgattccaaaaaaggggcatgggagaaaggatcgtttcctctcatcatcaataaacttaagatggaagactctcag acttatatctgtgagctggagaacaggaaagaggaggtggagttgtgggtgttcaaagtgaccttcagtccgggtaccagcctgttgc aagggcagagcctgaccctgaccttggatagcaactctaaggtctctaacccccttgacagagtgcaaacacaaaaagggtaaagttgt cagtggttccaaagttctctccatgtccaacctaagggttcaggacagcgacttctggaactgcaccgtgaccctggaccagaaaaag aactggttcggcatgacactctcagtgctgggttttcagagcacagctatcacggcctataagagtgagggagagtcagcggagttct ccttcccactcaactttgcagaggaaaacgggtggggagagctgatgtggaaggcagagaaggattctttcttccagccctggatctc cttctccataaagaacaaagaggtgtccgtacaaaagtccaccaaagacctcaagctccagctgaaggaaacgctcccactcaccctc aagataccccaggtctcgcttcagtttgctggttctggcaacctgactctgactctggacaaagggacactgcatcaggaagtgaacc tggtggtgatgaaagtggctcagctcaacaatactttgacctgtgaggtgatgggacctacctctcccaagatgagactgaccctgaa gcaggagaaccaggaggccagggtctctgaggagcagaaagtagttcaagtggtggcccctgagacagggctgtggcagtgtct actgagtgaaggtgataaggtcaagatggactccaggatccaggttttatccagaggggtgaaccagacagtgttcctggcttgcgtg ctg<u>ggt**GTG**tccttc**CTC**</u>tttctgggtttccttgggctctgcatcctctgctgtgtcaggtgccggtgataaagatcttgagtcgacaag cttgctag.

## T1-Palm (2C)

acggaattccgctcgagcgccaccatggtgcgagccatctctcttaggcgcttgctgctgctgctgcagctgtcacaactcctag ctgtcactcaagggaagacgctggtgctggggaaggaagggggaatcagcagaactgccctgcgagagttcccagaagaagatca cagtcttcacctggaagttctctgaccagaggaagattctggggcagcatggcaaaggtgtattaattagaggaggttcgccttcgca gtttgatcgtttttgattccaaaaaaggggcatgggagaaaggatcgtttcctctcatcatcaataaacttaagatggaagactctcagactt atatctgtgagctggagaacaggaaagaggaggtggagttgtgggtgttcaaagtgaccttcagtccgggtaccagcctgttgcaagg gcagagcctgaccctgaccttggatagcaactctaaggtctctaacccccttgacagagtgcaaacacaaaaagggtaaagttgtcagt ggttccaaagttctctccatgtccaacctaagggttcaggacagcgacttctggaactgcaccgtgaccctggaccagaaaaagaact ggttcggcatgacactctcagtgctgggttttcagagcacagctatcacggcctataagagtgagggagagtcagcggagttctcctt cccactcaactttgcagaggaaaacgggtggggagagctgatgtggaaggcagagaaggattctttcttccagccctggatctccttc tccataaagaacaaagaggtgtccgtacaaaagtccaccaaagacctcaagctccagctgaaggaaacgctcccactcaccctcaaga

taccccaggtctcgcttcagtttgctggttctggcaacctgactctgactctggacaaagggacactgcatcaggaagtgaacctggt
ggtgatgaaagtggctcagctcaacaatactttgacctgtgaggtgatgggacctacctctcccaagatgagactgaccctgaagcag
gagaaccaggaggccagggtctctgaggagcagaaagtagttcaagtggtggcccctgagacagggctgtggcagtgtctactg
agtgaaggtgataaggtcaagatggactccaggatccaggtttatccagagggtgaaccagacagtgttcctggcttgcgtgctgg
gtggctccttcggctttctgggtttccttgggctctgcatcctctgcTCTgtcaggTCCcggtgataaagatcttgagtcgacaagcttg
ctag.

## T1-Palm (3C)

acggaattccgctcgagcgccaccatggtgcgagccatctctcttaggcgcttgctgctgctgctgctgcagctgtcacaactcctag
ctgtcactcaagggaagacgctggtgctggggaaggaagggggaatcagcagaactgccctgcgagagttcccagaagaagatca
cagtcttcacctggaagttctctgaccagaggaagattctggggcagcatggcaaaggtgtattaattagaggaggttcgccttcgca
gtttgatcgtttttgattccaaaaaaggggcatgggagaaaggatcgtttcctctcatcatcaataaacttaagatggaagactctcagactt
atatctgtgagctggagaacaggaaagaggaggtggagttgtgggtgttcaaagtgaccttcagtccgggtaccagcctgttgcaagg
gcagagcctgaccctgaccttggatagcaactctaaggtctctaaccccttgacagagtgcaaacacaaaaagggtaaagttgtcagt
ggttccaaagttctctccatgtccaacctaagggttcaggacagcgacttctggaactgcaccgtgaccctggaccagaaaaagaact
ggttcggcatgacactctcagtgctgggttttcagagcacagctatcacggcctataagagtgagggagagtcagcggagttctcctt
cccactcaactttgcagaggaaacgggtggggagagctgatgtggaaggcagagaaggattctttcttccagccctggatctccttc
tccataaagaacaaagaggtgtccgtacaaaagtccaccaaagacctcaagctccagctgaaggaaacgctcccactcaccctcaaga
taccccaggtctcgcttcagtttgctggttctggcaacctgactctgactctggacaaagggacactgcatcaggaagtgaacctggt
ggtgatgaaagtggctcagctcaacaatactttgacctgtgaggtgatgggacctacctctcccaagatgagactgaccctgaagcag
gagaaccaggaggccagggtctctgaggagcagaaagtagttcaagtggtggcccctgagacagggctgtggcagtgtctactg
agtgaaggtgataaggtcaagatggactccaggatccaggtttatccagagggtgaaccagacagtgttcctggcttgcgtgctgg
gtggctccttcggctttctgggtttccttgggctctgcatcctcTCCTCTgtcaggTCCcggtgataaagatcttgagtcgacaagcttg
ctag.

## T1-TP (2C)

acggaattccgctcgagcgccaccatggtgcgagccatctctcttaggcgcttgctgctgctgctgctgcagctgtcacaactcctag
ctgtcactcaagggaagacgctggtgctggggaaggaagggggaatcagcagaactgccctgcgagagttcccagaagaagatca
cagtcttcacctggaagttctctgaccagaggaagattctggggcagcatggcaaaggtgtattaattagaggaggttcgccttcgca
gtttgatcgtttttgattccaaaaaaggggcatgggagaaaggatcgtttcctctcatcatcaataaacttaagatggaagactctcag
acttatatctgtgagctggagaacaggaaagaggaggtggagttgtgggtgttcaaagtgaccttcagtccgggtaccagcctgttgc
aagggcagagcctgaccctgaccttggatagcaactctaaggtctctaaccccttgacagagtgcaaacacaaaaagggtaaagttgt
cagtggttccaaagttctctccatgtccaacctaagggttcaggacagcgacttctggaactgcaccgtgaccctggaccagaaaaag
aactggttcggcatgacactctcagtgctgggttttcagagcacagctatcacggcctataagagtgagggagagtcagcggagttct
ccttcccactcaactttgcagaggaaacgggtggggagagctgatgtggaaggcagagaaggattctttcttccagccctggatctc
cttctccataaagaacaaagaggtgtccgtacaaaagtccaccaaagacctcaagctccagctgaaggaaacgctcccactcaccctc
aagataccccaggtctcgcttcagtttgctggttctggcaacctgactctgactctggacaaagggacactgcatcaggaagtgaacc
tggtggtgatgaaagtggctcagctcaacaatactttgacctgtgaggtgatgggacctacctctcccaagatgagactgaccctgaa
gcaggagaaccaggaggccagggtctctgaggagcagaaagtagttcaagtggtggcccctgagacagggctgtggcagtgtct
actgagtgaaggtgataaggtcaagatggactccaggatccaggtttatccagagggggtgaaccagacagtgttcctggcttgcgtg
ctgggtGTGtccttcCTCtttctgggtttccttgggctctgcatcctctgcTCTgtcaggTCCcggtgataaagatcttgagtcga
caagcttgctag.

## T1-TP (3C)

acggaattccgctcgagcgccaccatggtgcgagccatctctcttaggcgcttgctgctgctgctgctgcagctgtcacaactcctag
ctgtcactcaagggaagacgctggtgctggggaaggaagggggaatcagcagaactgccctgcgagagttcccagaagaagatca
cagtcttcacctggaagttctctgaccagaggaagattctggggcagcatggcaaaggtgtattaattagaggaggttcgccttcgca
gtttgatcgtttttgattccaaaaaaggggcatgggagaaaggatcgtttcctctcatcatcaataaacttaagatggaagactctcag
acttatatctgtgagctggagaacaggaaagaggaggtggagttgtgggtgttcaaagtgaccttcagtccgggtaccagcctgttgc
aagggcagagcctgaccctgaccttggatagcaactctaaggtctctaaccccttgacagagtgcaaacacaaaaagggtaaagttgt
cagtggttccaaagttctctccatgtccaacctaagggttcaggacagcgacttctggaactgcaccgtgaccctggaccagaaaaag
aactggttcggcatgacactctcagtgctgggttttcagagcacagctatcacggcctataagagtgagggagagtcagcggagttct
ccttcccactcaactttgcagaggaaacgggtggggagagctgatgtggaaggcagagaaggattctttcttccagccctggatctc
cttctccataaagaacaaagaggtgtccgtacaaaagtccaccaaagacctcaagctccagctgaaggaaacgctcccactcaccctc
aagataccccaggtctcgcttcagtttgctggttctggcaacctgactctgactctggacaaagggacactgcatcaggaagtgaacc

tggtggtgatgaaagtggctcagctcaacaatactttgacctgtgaggtgatgggacctacctctcccaagatgagactgaccctgaa
gcaggagaaccaggaggccagggtctctgaggagcagaaagtagttcaagtggtggcccctgagacagggctgtggcagtgtct
actgagtgaaggtgataaggtcaagatggactccaggatccaggtttatccagaggggtgaaccagacagtgttcctggcttgcgtg
ctgggtGTGtccttcCTCtttctgggtttccttgggctctgcatcctcTCCTCTgtcaggTCCcggtgataaagatcttgagtcga
caagcttgctag.

The gene of T1^Δbind was made by cloning the in-frame extracellular domain of CD4 that encodes the Δbind extracellular domain (*Parrish et al., 2015*) flanked by XhoI and BamHI into the T1 gene. The gene was sequence-verified in pUC 18, subcloned into pP2 puromycin resistance MSCV vector (MCS-IRES-puro) via 5' XhoI and 3' BglII, midi-prepped, and sequence-verified. The sequence for the construct is as follows with mutated codons shown in bold uppercase letters, while the motif under interrogation is shown in bold and underlined (any non-mutated codons within the motif are bold lowercase letters and underlined):

**T1^Δbind**

acggaattccgctcgagcgccaccatggtgcgagccatctctcttaggcgcttgctgctgctgctgctgcagctgtcacaactcctagctgt
cactcaagggaagacgctggtgctggggaaggaaggggaatcagcagaactgccctgcgagagttcccagaagaagatcacagt
cttcacctggaagttctctgaccagaggaagattctggggcagcatggc**GAT**ggt**GATTCAGATAGC**ggaggttcgccttcgc
agtttgatcgtttttgattccaaaaaaggggcatgggagaaaggatcgtttcctctcatcatcaataaacttaagatggaagactctcagact
tatatctgtgagctggagaacaggaaagaggaggtggagttgtgggtgttcaaagtgaccttcagtccgggtaccagcctgttgcaag
ggcagagcctgaccctgaccttggatagcaactctaaggtctctaaccccttgacagagtgcaaacacaaaaagggtaaagttgtcag
tggttccaaagttctctccatgtccaacctaagggttcaggacagcgacttctggaactgcaccgtgaccctggaccagaaaaagaac
tggttcggcatgacactctcagtgctgggttttcagagcacagctatcacggcctataagagtgagggagagtcagcggagttctcct
tcccactcaactttgcagaggaaacgggtggggagagctgatgtggaaggcagagaaggattctttcttccagccctggatctcctt
ctccataaagaacaaagaggtgtccgtacaaaagtccaccaaagacctcaagctccagctgaaggaaacgctcccactcaccctcaag
ataccccaggtctcgcttcagtttgctggttctggcaacctgactctgactctggacaaagggacactgcatcaggaagtgaacctgg
tggtgatgaaagtggctcagctcaacaatactttgacctgtgaggtgatgggacctacctctcccaagatgagactgaccctgaagca
ggagaaccaggaggccagggtctctgaggagcagaaagtagttcaagtggtggcccctgagacagggctgtggcagtgtctact
gagtgaaggtgataaggtcaagatggactccaggatccaggtttatccagaggggtgaaccagacagtgttcctggcttgcgtgctg
ggtggctccttcggctttctgggtttccttgggctctgcatcctctgctgtgtcaggtgccggtgataaagatcttgagtcgacaagcttgc
tag.

## Materials availability statement

All data generated and analyzed for inclusion in this study are included in the manuscript and supporting files. These include alignments and phylogenetic trees associated with *Figure 1*, as well as the plotted values and associated statistical values shown in figures (see Dryad: http://doi.org/10.5061/dryad.p8cz8w9vw). Cell lines and constructs are available upon request. All sequences are provided in the Materials and methods.

## Acknowledgements

This work was supported by the National Institutes of Health/National Institute of Allergy and Infectious Diseases Grant R01 AI101053 (MSK), the Cancer Center Support Grant CCSG-CA 023074, and AZ TRIF funds from the BIO5 Institute (KVD). We thank Thomas Serwold and Dominik Schenten for critical feedback on the manuscript.

## Additional information

### Competing interests

Michael S Kuhns: Has disclosed an outside interest in Module Therapeutics to the University of Arizona. Conflicts of interest resulting from this interest are being managed by the University of Arizona in accordance with their policies. The other authors declare that no competing interests exist.

## Funding

| Funder | Grant reference number | Author |
|---|---|---|
| National Institute of Allergy and Infectious Diseases | R01 AI101053 | Michael S Kuhns |
| National Cancer Institute | CCSG-CA 023074 | Michael S Kuhns |
| BIO5 Institute, University of Arizona | | Koenraad Van Doorslaer |

The funders had no role in study design, data collection and interpretation, or the decision to submit the work for publication.

## Author contributions

Mark S Lee, Conceptualization, Data curation, Formal analysis, Investigation, Methodology, Writing – review and editing; Peter J Tuohy, Caleb Y Kim, Katrina Lichauco, Data curation, Formal analysis, Investigation, Methodology, Writing – review and editing; Philip P Yost, Data curation, Formal analysis, Investigation, Writing – review and editing; Heather L Parrish, Conceptualization, Methodology, Writing – review and editing; Koenraad Van Doorslaer, Conceptualization, Data curation, Formal analysis, Supervision, Funding acquisition, Validation, Visualization, Methodology, Writing – original draft, Writing – review and editing; Michael S Kuhns, Conceptualization, Data curation, Supervision, Funding acquisition, Methodology, Writing – original draft, Project administration, Writing – review and editing

## Author ORCIDs

Caleb Y Kim (ORCID) http://orcid.org/0000-0003-0093-3049
Katrina Lichauco (ORCID) http://orcid.org/0000-0002-9480-2893
Koenraad Van Doorslaer (ORCID) http://orcid.org/0000-0002-2985-0733
Michael S Kuhns (ORCID) https://orcid.org/0000-0002-0403-6313

Reviewer #1 (Public Review): https://doi.org/10.7554/eLife.88225.3.sa1
Reviewer #2 (Public Review): https://doi.org/10.7554/eLife.88225.3.sa2
Author response https://doi.org/10.7554/eLife.88225.3.sa3

# Additional files

## Supplementary files

• MDAR checklist

## Data availability

Raw data, include alignments and phylogenetic trees associated with *Figure 1* as well as the source data associated statistical values shown in remaining figures, are available on Dryad (see Dryad: https://doi.org/10.5061/dryad.p8cz8w9vw).

The following previously published dataset was used:

| Author(s) | Year | Dataset title | Dataset URL | Database and Identifier |
|---|---|---|---|---|
| Lee M, Tuohy P, Kim C, Yost P, Lichauco K, Parrish H, Van Doorslaer K, Kuhns M | 2024 | Data for: The CD4 transmembrane GGXXG and juxtamembrane (C/F) CV+C motifs mediate pMHCII-specific signaling independently of CD4-Lck interactions | https://doi.org/10.5061/dryad.p8cz8w9vw | Dryad Digital Repository, 10.5061/dryad.p8cz8w9vw |

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
