## [Editor Report · eLife assessment]

This study provides **valuable** new insights as to how two evolutionary conserved motifs in CD4 contribute to the CD4-mediated enhancement of TCR signaling independently of the CD4-LCK interaction. The data at hand are **convincing**, even if confined to a cell line model and not substantiated in vivo and with little new mechanistic insight provided regarding the domains of CD4 shown to have significant roles in the signaling process. Without the data from primary cells it is difficult to make statements about the quantitative contribution of LCK-dependent and independent functions of CD4 in TCR signaling.

---

## [Referee Report · Reviewer #1 (Public Review)]

Summary:

This study by Lee et al. is a direct follow-up on their previous study that described an evolutionary conservancy among placental mammals of two motifs (a transmembrane motif and a juxtamembrane palmitoylation site) in CD4, an antigen co-receptor, and showed their relevance for T-cell antigen signaling. In this study, they describe the contribution of these two motifs to the CD4-mediated antigen signaling in the absence of CD4-LCK binding. Their approach was the comparison of antigen-induced proximal TCR signaling and distal IL-2 production in 58-/- T-cell hybridoma expressing exogenous truncated version of CD4 (without the interaction with LCK), called T1 and T1 version with the mutations in either or both of the conserved motifs. They show that the T1 CD4 can support signaling to extend similar to WT CD4, but the mutation of the conserved motifs substantially reduced the signaling. The authors conclude that the role of these motifs is independent of the LCK-binding.

Strengths:

The authors convincingly show that CD4 is capable of contributing to TCR signaling in a manner independent of LCK, but dependent on the two studied motifs in CD4.

Weaknesses:

(1) Experiments in primary T cells are required to estimate the relative contribution of LCK-dependent and LCK-independent mechanisms of CD4 signaling.

(2) The mechanistic explanation (beyond the independence of LCK binding) of the role of these motifs is unclear at the moment.

---

## [Referee Report · Reviewer #2 (Public Review)]

Summary:

The paper by Kuhn and colleagues follows upon a 2022 eLife paper in which they identified residues in CD4 constrained by evolutionary purifying selection in placental mammals, and then performed functional analyses of these conserved sequences. They showed that sequences distinct from the CXC "clamp" involved in recruitment of Lck have critical roles in TCR signaling, and these include a glycine-rich motif in the transmembrane (TM) domain and the cys-containing juxtamembrane (JM) motif that undergoes palmitylation, both of which promote TCR signaling, and a cytoplasmic domain helical motif, also involved in Lck binding, that constrains signaling. Mutations in the transmembrane and juxtamembrane sequences led to reduced proximal signaling and IL-2 production in a hybridoma's response to antigen presentation, despite retention of abundant CD4 association with Lck in the detergent-soluble membrane fraction, presumably mislocalized outside of lipid rafts and distal to the TCR. A major conclusion of that study was that CD4 sequences required for Lck association, including the CXC "clasp" motif, are not as consequential for CD4 co-receptor function in TCR signaling as the conserved TM and JM motifs. However, the experiments did not determine whether the functions of the TM and JM motifs are dependent on the Lck-binding properties of CD4 - the mutations in those motifs could result in free Lck redistributing to associate with CD4 in signaling-incompetent membrane domains or could function independently of CD4-Lck association. The current study addresses this specific question.

Using the same model system as in the earlier eLife paper (the entire methods section is a citation to the earlier paper), the authors show that truncation of the Lck-binding intracellular domain resulted in a moderate reduction in IL-2 response, as previously shown, but there was no apparent effect on proximal phosphorylation events (CD3z, Lck, ZAP70, PLCg1). They then evaluated a series of TM and JM motif mutations in the context of the truncated Lck-nonbinding molecule and showed that these had substantially impaired co-receptor function in the IL-2 assay and reduced proximal signaling. The proximal signaling could be observed at high ligand density even with a MHC non-binding mutation in CD4, although there was still impaired IL-2 production. This result additionally illustrates that phosphorylation of the proximal signaling molecules is not sufficient to activate IL-2 expression in the context of antigen presentation.

Strengths:

The strength of the paper is the further clear demonstration that the classical model of CD4 co-receptor function (MHCII-binding CD4 bringing Lck to the TCR complex, for phosphorylation of the CD3 chain ITAMs and of the ZAP70 kinase) is not sufficient to explain TCR activation. The data, combined with the earlier eLife paper, further implicate the gly-rich TM sequence and the palmitylation targets in the JM region as having critical roles in productive co-receptor-dependent TCR activation.

Weaknesses:

The major weakness of the paper is the lack of mechanistic insight into how the TM and JM motifs function. The new results are largely incremental in light of the earlier paper from this group as well as other literature, cited by the authors, that implicates "free" Lck, not associated with co-receptors, as having the major role in TCR activation. It is clear that the two motifs are important for CD4 function at low pMHCII ligand density. The proposal that they modulate interactions of TCR complex with cholesterol or other membrane lipids is an interesting one, and it would be worth further exploring by employing approaches that alter membrane lipid composition. The JM sequence presumably dictates localization within the membrane, by way of palmitylation, which may be critical to regulate avidity of the TCR:CD4 complex for pMHCII or TCR complex allosteric effects that influence the activation threshold. Experiments that explore the basis of the mutant phenotype could substantially enhance the impact of this study.

Additional comments:

- Is the "IL-2 sensitivity" measurement for the T1-TP (3C) meaningful (Table 3)? It is showing only a moderate reduction compared to T1 control, while TP (2C) or just the 3C palmitylation mutations essentially eliminate response.

- It is unclear how the pairs of control and mutant cells connected by lines in the figures are related. They are presumably cells from distinct biological experiments, with technical replicates for each, but are they paired because they were derived at the same time with different constructs? This should be explained in this paper, not in a reference.

---

## [Author Response]

The following is the authors’ response to the previous reviews.

**Reviewer #1 (Public Review):**
Summary:This study by Lee et al. is a direct follow-up on their previous study that described an evoluBonary conservancy among placental mammals of two moBfs (a transmembrane moBf and a juxtamembrane palmitoylaBon site) in CD4, an anBgen co-receptor, and showed their relevance for T-cell anBgen signaling. In this study, they describe the contribuBon of these two moBfs to the CD4-mediated anBgen signaling in the absence of CD4-LCK binding. Their approach was the comparison of anBgen-induced proximal TCR signaling and distal IL-2 producBon in 58-/- T-cell hybridoma expressing exogenous truncated version of CD4 (without the interacBon with LCK), called T1 with T1 version with the mutaBons in either or both of the conserved moBfs. They show that the T1 CD4 can support signaling to the extend similar to WT CD4, but the mutaBon of the conserved moBfs substanBally reduced the signaling. The authors conclude that the role of these moBfs is independent of the LCK-binding.Strengths:The authors convincingly show that T1 CD4, lacking the interacBon with LCK supports the TCR signaling and also that the two studied moBfs have a significant contribuBon to it.Weaknesses:The study has several weaknesses.(1) The whole study is based on a single experimental system, geneBcally modified 58-/- hybridoma. It is unclear at this moment, how the molecular moBfs studied here contribute to the signaling in a real T cell. The evoluBonary conservancy suggests that these moBfs are important for T cell biology. However, the LCK-binding moBf is conserved as well (perhaps even more) and it plays a very minor role in their model. Without verifying their results in primary cells, the quanBtaBve, but even qualitaBve, importance of these moBfs for T-cell signaling and biology is unclear. Although the authors discuss this issue in the Discussion, it should be noted in all important parts of the manuscript, where conclusions are made (abstract, end of introducBon, perhaps also in the Btle) that the results are coming from the hybridoma cells.

We appreciate the Reviewer’s thoughWul comments and suggesBon. We now state in the abstract and introducBon that wet-lab experiments were performed with T cell hybridomas. We have also beXer highlighted work from Killeen and LiXman (PMID: 8355789) wherein they showed that C-terminally truncated CD4, which lacked the moBfs that mediate CD4-Lck interacBons, can drive CD4+ T cell development, proliferaBon, and T-helper funcBon because we now provide mechanisBc data to help explain those in vivo results. Also, as noted by the reviewer, we discuss how the sum of our data provides jusBficaBon for the investment in and use of mouse models to interrogate how the funcBonally important residues/moBfs idenBfied and studied here influence T cell biology.

We will take the opportunity to reiterate here that, while the study is based on a well characterized, albeit single, wet-lab experimental system, the whole study is based on two lines of invesBgaBon. The other approach was a systems biology computaBonal approach that analyzes data from real-world experiments in a variety of jawed vertebrate species over evoluBon. Specifically, we used a computaBonal reconstrucBon of the evoluBonary history of CD4 by performing mulBple analyses of CD4 from 99 jawed vertebrates spanning ~435 million years of evoluBon. This analysis allowed us to idenBfy residues, and networks of evoluBonarily coupled residues, that are predicted to be funcBonally important in vivo. Like other systems biology approaches, this allowed us to look at the larger picture by evaluaBng data points that have emerged from constant tesBng and adjustments of CD4 funcBon in vivo through selecBon on an evoluBonary Bmescale in more jawed vertebrate species, and under more real-world condiBons, than can be tested in the laboratory. Our structure-funcBon analysis provided a second, wet-lab reducBonist experimental system to cross-validate that the residues idenBfied by our evoluBonary analysis are funcBonally significant. This experimental validaBon is criBcal and elevates the relevance of our studies above ad hoc observaBons. Our work also provides mechanisBc insights for why the residues studied here are funcBonally significant (i.e., key determinants of pMHCII-specific signaling iniBaBon). In short, using both systems allowed us to cross-validate the funcBonal significance of the residues within the GGXXG and (C/F)CV+C moBfs studied here by two independent methods.

(2) Many of the experiments lack the negaBve control. I believe that two types of negaBve controls should be included in all experiments. First, hybridoma cells without CD4 (or with CD4 mutant unable to bind MHCII). Second, no pepBde control, i.e., acBvaBon of the hybridoma cells with the APC not loaded with the cognate pepBde. These controls are required to disBnguish the basal levels of phoshorylaBon and CD4-independent anBgen-induced phosphorylaBon to quanBfy, what is the contribuBon of the parBcular moBfs to the CD4-mediated support. Although these controls are included in some of the experiments, they are missing in other ones. The binding mutant appears in some FC results as a horizontal bar (without any error bar/variability), showing that CD4 does not give a huge advantage in these readouts. Why don't the authors show no pepBde controls here as well? Why the primary FC data (histograms) are not shown? Why neither of these two controls is shown for the % of responders plots? Although the IL-2 producBon is a very robust and convincing readout, the phosphoflow is much less sensiBve. It seems that the signaling is elevated only marginally. Without the menBoned controls and showing the raw data, the precise interpretaBon is not possible.

These comments, and those in point #3, concern our flow cytometry-based analysis of early intracellular signaling events where we asked: how do the moBfs under invesBgaBon impact phosphorylaBon of CD3z, ZAP-70, and PLCg1 in response to agonist pMHCII? Thank you for poinBng out areas of confusion regarding these analyses. We will try to clarify here and have worked to clarify the text.

Our approach was to mutate consBtuent residues within the moBfs that our evoluBonary analysis predicted to be funcBonally significant, compare the performance of the mutants to that of controls bearing WT moBfs, and then infer the funcBon of the moBfs based on the differenBal phenotype of the mutants relaBve to their controls. In most cases, the C-terminally truncated CD4-T1 mutant served as the appropriate CD4 control backbone against which to evaluate the phenotypes of the GGXXG and (C/F)CV+C moBf mutants. This is a convenBonal structure-funcBon strategy.

All experiments included APCs expressing null pMHCII (Hb:I-Ek) as negaBve controls. These were a necessary component of the data analysis, explained further below, which involved background subtracBon of the signal from control or mutant T cell hybridomas bound to these negaBve control APCs from those bound to the agonist pMHCII (MCC:I-Ek). Doing so allowed us to establish a true signal over background for calculaBng percent responders and signaling intensity. These negaBve controls served the same purpose of APCs expressing I-Ek not loaded with cognate pepBde requested by the reviewer. It is important to note that we previously published that TCR-CD3-pMHCII interacBons reciprocally increase CD4-pMHCII dwell Bme, and vice versa, such that dwell Bmes of the 5c.c7 TCR and CD4 to the null Hb:I-Ek are both basal in this system relaBve to antagonist, weak agonist, and agonist pMHCII (PMID 29386113). A recent study using different techniques also concluded that TCR-CD3 and CD4 cooperaBvely enhance signaling to pMHCII (PMID 36396644). The use of the null pMHCII, Hb:I-Ek, in each experiment thus serves as a well-characterized negaBve control for both TCR and CD4 engagement in this experimental system with regards to assembly of the TCR-CD3 and CD4 around pMHCII to drive signaling. In our view, it is the most important negaBve control for interpreBng our results, and it is present in each experiment. In Fig 1B and related supplemental figures we compare the Cterminally truncated CD4-T1 mutant to the full-length WT CD4 to evaluate the contribuBons of the intracellular domains to early signaling events. We found no significant differences for pCD3z, pZAP-70, and pPLCg1 levels demonstraBng that, in our system, CD4 WT and T1 are staBsBcally indisBnguishable.

In Fig 1C we asked: what is the contribuBon of CD4-pMHCII interacBons made by CD4 T1, which lacks the intracellular domain, using our CD4 T1Dbind mutant. Fig 2C and Table 3 show that pCD3z levels for T1Dbind were ~54% of T1, meaning that CD4 binding to pMHCII roughly doubles pCD3z levels (even without the intracellular domain). We also showed that the percent of responders were not different between the CD4 T1 and T1Dbind mutant in Fig 2C. The impact on ZAP-70 and PLCg1 are shown in Figure 2—figure supplement 4. These differences, including the magnitude of the decrease, were observed reproducibly (p<0.001) in three independently generated sets of lines. We believe that this analysis saBsfies the request by the reviewer for an analysis of the contribuBons of CD4 binding to pMHCII. We did not include this as a negaBve control in experiments evaluaBng the contribuBons of the GGXXG and (C/F)CV+C moBfs to CD4 T1 signaling because the quesBon being asked in those experiments was how do the moBfs impact signaling in the absence of the intracellular domain (i.e., within the CD4 T1 backbone, making CD4 T1 the proper comparator for the quesBon we were asking). We showed the average normalized intensity for the T1Dbind mutant, relaBve to T1, for this lower bound of signaling mediated by TCR-CD3-only as a doXed line in those figures to provide a reference point for the readers to evaluate and put into perspecBve how the mutants we generated impacted the overall contribuBon of CD4 to these early signaling events. The T1Dbind mutants were not always measured in the same experiment at the same Bme with other mutants, because the cell lines used were not always made at the same Bme, so we did not think it appropriate to graph the results together.

We do not know how to interpret the comment “Although the IL-2 producBon is a very robust and convincing readout, the phosphoflow is much less sensiBve. It seems that the signaling is elevated only marginally.” We will offer our perspecBve that we do not know how to equate the sensiBvity of the phos-flow to the IL-2. Because the IL-2 is a signaling output, it results from signaling amplificaBon from the membrane to the nucleus. If CD3z phosphorylaBon is the iniBaBng event for a signaling cascade that leads to IL-2 gene transcripBon and transducBon, as is widely believed, our data strongly suggests that the ~2-fold difference in pCD3z levels between CD4 T1 and T1Dbind (Fig 2C/Table 3 data) contributes to the difference between no IL-2 output for T1Dbind and IL-2 output by T1 in this experimental system. Because CD4 WT and T1 have significantly different levels of IL-2 output, but show no significant differences in pCD3z, pZAP-70, or pPLCg1 levels, there are likely to be other differences we did not measure via other pathways that intersect at the nucleus. At many levels, biology works on gradients such that small differences can Bp a system in one direcBon or another. The kineBc discriminaBon model (PMID 8643643), which is thought to be a reasonable descripBon of the relaBonship between pMHC engagement and signaling outcomes, suggests that very small differences in molecular interacBons at the earliest stages of a response can lead to big differences in signaling outcome. We therefore have no basis at this juncture to think that ~2-fold differences in pCD3z levels could not account for bigger differences in signaling output such as IL-2.

(3) The processing of the data is not clear. Some of the figures seem to be overprocessed. For instance, I am not sure what "Normalized % responders of pCD3zeta" means (e.g., Fig. 1C and elsewhere)? Why do not the authors show the actual % of pCD3zeta+ cells including the gaBng strategy? Why do the authors subtract the two histograms in Fig. 2- Fig.S3? It is very unusual.

We did develop and implement a novel strategy for measuring the impact of our mutaBons onCD3z, ZAP-70, and PLCg1 phosphorylaBon. This was explained in more detail in our prior study. The instrucBons to authors indicated that we should not repeat methods in the current manuscript. However, we will go through the approach here, and address why we did not show primary FC histograms for all experiments from above. First, we think that a brief explanaBon as to what moBvated us to develop our approach will add to a beXer understanding:

(1) For experimental and staBsBcal rigor, our goal was to perform both experimental and biological replicates by measuring and comparing the average of at least three independently generated sets of paired WT/T1 control Vs. mutant cells lines generated at different Bmes to determine the staBsBcal significance of the difference, if any, between averages of the control and mutant lines.

(2) Our quesBons necessitated that we measure signals generated naturally by the cooperaBve engagement of cognate pMHCII by TCR-CD3 and CD4 on APCs, rather than through aCD3/aCD4 crosslinking.

(3) We chose to use flow cytometry rather than bulk cell analysis by Western Bloung to analyze signaling occurring in cells that were engaged to the agonist APC in order to avoid diluBon of that signal by cells that are not engaged to APCs and not signaling. 4. For each experiment, we wanted to subtract background signals from cells bound to APCs expressing a null pMHCII (Hb:I-Ek) from signals generated by cells bound to APCs expressing agonist pMHCII (MCC:I-Ek). Doing so allowed us to idenBfy cells that are signaling (responders) to agonist over null pMHCII. The goal here was to quanBtate the level of signaling in an objecBve manner with a method that can be applied to all samples uniformly rather than seung a flow cytometry gate on posiBve cells (e.g. pCD3z) because gaBng is subjecBve and can vary from experiment to experiment. To put that another way, as detailed below, we used our subtracBon method to idenBfy signaling responders rather than seung a signaling gate on the posiBve populaBon.

Regarding gaBng schemes, controls, and data processing:

Figure 2—figure supplement 3 of the current study and Figure 6—figure supplement 1 of our prior study are designed to walk the reader through our experimental design, gaBng, data processing and thinking. Here we will provide a detailed explanaBon to complement the figure legend as well as the methods provided in our prior manuscript (see pt #4 below).

We will refer to Figure 2—figure supplement 3 here:

Panel A. The dot plots show our approach to idenBfying 5c.c7+ CD4+ 58a-b- T cell hybridomas (yaxis, GFP posiBve) coupled to M12 cells (x-axis, TagIt Violet) expressing the null pMHCII Hb:I-Ek (lev) or agonist pMHCII MCC:I-Ek (right). The gaBng shows the frequency of GFP+ T cell hybridomas that are bound to TagIt violet posiBve APCs (i.e., cell couples). The histogram on the right then shows the staining intensity for pCD3z on the x-axis for the 10,000 coupled events collected wherein the APCs express the null pMHCII (filled cyan) or the agonist pMHCII (black line).

Panel B. The data presented here is the same as in Panel A, but for CD4 T1 cells.

Panel C. The data presented here walks through how we idenBfy 5c.c7+ CD4+ 58a-b- T cell hybridomas responding (i.e., signaling) to agonist pMHCII, as well as the mean signaling intensity of the responding populaBon, in a gaBng-independent manner aver background subtracBon. For the lev graph, we exported the data for the histograms shown in Panel A from FlowJo 10 sovware and ploXed them here using Prism 9 as smoothed lines (500 nearest neighbors). The cyan line is therefore a replicate of the flow cytometry histogram shown in Panel A for pCD3z intensity from 5c.c7+ CD4+ 58a-b- T cell hybridomas coupled to M12 cells expressing the null pMHCII (Hb:I-Ek), while the black histogram is a replicate of the pCD3z intensity for 5c.c7+ CD4+ 58a-b- T cell hybridomas coupled to M12 cells expressing the agonist pMHCII (MCC:I-Ek). Next, to idenBfy the responding populaBon in a gaBng-independent manner, we used Excel to subtract the pCD3z intensity for the null pMHCII (cyan) negaBve control populaBon on a bin-by-bin bases from the pCD3z intensity for the agonist pMHCII (black) responding populaBon. We then transferred the background subtracted values to Prism 9 for smoothing and ploung (grey line: MCC:I-Ek minus Hb:I-Ek). The middle graph shows the same data processing for the data from Panel B for the CD4 T1 cells. Please note that the background subtracted grey line has negaBve values and posiBve values. The negaBve values represent intensity bins where signaling in response to agonist pMHCII leads to fewer cells per bin than in the null pMHCII populaBon that is not signaling, while the posiBve values represent bins of intensity where signaling cells outnumber non-signaling cells. The right graph in this panel shows the populaBons aver background subtracBon for intensity bins that had more cells with pCD3z signal in the agonist pMHCII populaBon than the null pMHCII populaBon (grey = WT full length CD4 and blue = T1). In short, the right graph shows idenBficaBon of those cells that are signaling in response to agonist pMHCII. This approach miBgated the need for subjecBve gaBng in FlowJo to idenBfy signaling cells (i.e., pCD3z posiBve) and allowed for background subtracBon which could not be done in FlowJo. We used this approach for all analyses of pCD3z, pZAP-70, and pPLCg1 in this study.

The number of cells in these background-subtracted populaBons were divided by 10,000 (the number of events collected and analyzed) to calculate the percent of responding 5c.c7+ CD4+ 58a-b- T cell hybridomas, while the mean fluorescent intensity for the cells within these populaBon represent the signaling intensity.

Panel D. The graph on the lev shows the mean fluorescence intensity (MFI) ± SEM for the posiBve signaling populaBon from the right graph of panel C. We see in this example comparing a WT and T1 cell line, generated at the same Bme from the same parental 58a-b- T cell hybridoma populaBon, that the T1 MFI is significantly greater than the WT. These intensity values represent one of the paired intensity values used in the main Fig 2B (Lev graph), where we show the paired MFI analysis of responding populaBons from 5 independently generated sets of cell lines. Please note that these single MFI values are directly derived from the flow cytometry histograms aver background subtracBon. Figure 2B, and similar figures, therefore equate to a disBllaBon of all of the histograms for the populaBons tested in a manner that we consider easier to digest than either overlaying all histograms or showing mulBple panels individually. It also conserves more space. This is why we only showed representaBve flow cytometry histograms, rather than all histograms.

The graph on the right shows the % responders for the posiBve signaling populaBon from the right graph of panel C. Specifically, the total number of cells that were determined to be signaling in response to agonist pMHCII was divided by 10,000 (the number of coupled cells collected by flow cytometry) to determine the percent responders. These values represent one of five sets of values used to determine the average normalized percent responders (all normalized to WT). There was no significant difference between these two populaBons in terms of percent responders.

Regarding graphing normalized values for the mean MFI for signaling intensity or the percent responders: in our first manuscript, we presented the individual MFI intensity values for matched pairs of cells as well as the actual percent responders per group. The feedback we received from colleagues on this presentaBon was that it was confusing, distracBng, and otherwise hard to digest. It was suggested to us by mulBple individuals that the normalized values would be preferable because it is easier and faster to understand. Upon reflecBon, we agreed with this feedback because the normalized presentaBon with staBsBcs allows for the two key relevant quesBons to be quickly evaluated: 1. Are the mutants different than the control? 2. By how much? We have lev the raw intensity values and well as the normalized intensity values in the version of record. Given the Reviewer’s comments, we have now graphed the average % responders instead of normalized values in the figures, and lev the normalized values in Table 3.

(4) The manuscript lacks Materials and Methods. It only refers to the previous paper, which is very unusual. Although most of the methods are the same, they sBll should be menBoned here. Moreover, some of the mutants presented here were not generated in the previous study, as far as I understand. Perhaps the authors plan to include Materials and Methods during the revision...

Because we submiXed this as a Research Advances arBcle we followed the journal instrucBons to reference the Materials and Methods in our prior publicaBon, upon which this work builds, as the methods used are the same. They are detailed in that study. We have now included a copy of the Materials and Methods for the eLife staff to determine how best to link with this manuscript. We have also included the gene sequences for the novel constructs used in this study. Thank you for poinBng out the omission.

(5) Membrane rafts are a very controversial topic. I recommend the authors stick to the more consensual term "detergent resistant microdomains" in all cases/occurances.

We agree this is a controversial topic with a variety of viewpoints. Because we are not experts in the field of membrane composition, we turned to the literature to inform our view of how best to refer to these membrane subdomains. In our reading, we found a 2006 meeting report from a Keystone symposium on lipid rafts and cell function authored by Linda Pike (PMID 16645198).At this meeting, a central focus was reaching a consensus on how best to refer to these domains. The consensus term agreed upon by this group was “membrane rafts”. Specifically, we will quote from this report published in the Journal of Lipid Research, ‘Together, the discussions permitted the generation of a definition for “lipid rafts” in an ad hoc session on the final day of the meeting. All participants were invited to contribute to this effort, and the work product reflects the consensus of this broad-based group…… First and foremost, the term “lipid raft” was discarded in favor of the term “membrane raft.”’ We chose to use the term “membrane raft” based on this consensus opinion.

(6) Last, but not least, the mechanistic explanation (beyond the independence of LCK binding) of the role of these motifs is very unclear at the moment.

We agree with this comment. One goal in making these results, and those in our prior study, available to the field at large is to provide evidence in support of our view that the dominant paradigm that is thought to explain the earliest events in T cell signaling needs re-evaluating. How T cell signaling is initiated in response to pMHCII is clearly more complex than is currently thought. However, out data is inconsistent with the dominant paradigm in which CD4 recruits Lck to TCR-CD3 to phosphorylate ITAMs to initiate signaling.

**Reviewer #2 (Public Review):**
Summary:The paper by Kuhn and colleagues follows upon a 2022 eLife paper in which they identified residues in CD4 constrained by evolutionary purifying selection in placental mammals and then performed functional analyses of these conserved sequences. They showed that sequences distinct from the CXC "clamp" involved in recruitment of Lck have critical roles in TCR signaling, and these include a glycine-rich motif in the transmembrane (TM) domain and the cyscontaining juxtamembrane (JM) motif that undergoes palmitoylation, both of which promote TCR signaling, and a cytoplasmic domain helical motif, also involved in Lck binding, that constrains signaling. Mutations in the transmembrane and juxtamembrane sequences led to reduced proximal signaling and IL-2 production in a hybridoma's response to antigen presentation, despite retention of abundant CD4 association with Lck in the detergent-soluble membrane fraction, presumably mislocalized outside of lipid rafts and distal to the TCR. A major conclusion of that study was that CD4 sequences required for Lck association, including the CXC "clasp" motif, are not as consequential for CD4 co-receptor function in TCR signaling as the conserved TM and JM motifs. However, the experiments did not determine whether the functions of the TM and JM motifs are dependent on the Lck-binding properties of CD4 - the mutations in those motifs could result in free Lck redistributing to associate with CD4 in signaling-incompetent membrane domains or could function independently of CD4-Lck association. The current study addresses this specific question.Using the same model system as in the earlier eLife paper (the entire methods section is a citation to the earlier paper), the authors show that truncation of the Lck-binding intracellular domain resulted in a moderate reduction in IL-2 response, as previously shown, but there was no apparent effect on proximal phosphorylation events (CD3z, Lck, ZAP70, PLCg1). They then evaluated a series of TM and JM motif mutations in the context of the truncated Lck-nonbinding molecule, and showed that these had substantially impaired co-receptor function in the IL-2 assay and reduced proximal signaling. The proximal signaling could be observed at high ligand density even with a MHC non-binding mutation in CD4, although there was still impaired IL-2 production. This result additionally illustrates that phosphorylation of the proximal signaling molecules is not sufficient to activate IL-2 expression in the context of antigen presentation.Strengths:The strength of the paper is the further clear demonstration that the classical model of CD4 coreceptor function (MHCII-binding CD4 bringing Lck to the TCR complex, for phosphorylation of the CD3 chain ITAMs and of the ZAP70 kinase) is not sufficient to explain TCR activation. The data, combined with the earlier eLife paper, further implicate the gly-rich TM sequence and the palmitoylation targets in the JM region as having critical roles in productive co-receptordependent TCR activation.Weaknesses:The major weakness of the paper is the lack of mechanistic insight into how the TM and JM motifs function. The new results are largely incremental in light of the earlier paper from this group as well as other literature, cited by the authors, that implicates "free" Lck, not associated with co-receptors, as having the major role in TCR activation. It is clear that the two motifs are important for CD4 function at low pMHCII ligand density. The proposal that they modulate interactions of TCR complex with cholesterol or other membrane lipids is an interesting one, and it would be worth further exploring by employing approaches that alter membrane lipid composition. The JM sequence presumably dictates localization within the membrane, by way of palmitoylation, which may be critical to regulate avidity of the TCR:CD4 complex for pMHCII or TCR complex allosteric effects that influence the activation threshold. Experiments that explore the basis of the mutant phenotype could substantially enhance the impact of this study.

We appreciate these thoughtful comments and suggestions. We will restate what we wrote in our preliminary response to the reviews to explain the scope of the current study:

To address comments about the limited scope of this study and referencing of the Methods secBon to our prior study, we would like to note that we submiXed the current study via the Research Advance mechanism. Our goal was to build upon the conclusions of our 2022 eLife publicaBon (PMID: 35861317) and address an unresolved quesBon from that study (as nicely summarized by Reviewer #2). In the current manuscript we present data from reducBonist experiments that were designed specifically for this purpose and, as noted by the reviewers, we provide answers to the quesBon being asked. We think that the Research Advance mechanism is an ideal opportunity to make these results available to the field given the stated purpose of such arBcles (for reference: “A Research Advance might use a new technique or a different experimental design to generate results that build upon the conclusions of the original research by, for example, providing new mechanis=c insights or extend the pathway under inves=ga=on…”).Now that we have provided evidence that CD4 does not recruit Lck to phosphorylate TCR-CD3 ITAMs in our system, nor do the GGXXG and (C/F)CV+C motifs play a role in enabling CD4 to regulate Lck proximity to TCR-CD3, we agree that it is important to form and test alternative hypotheses for how TCR-CD3 signaling is initiated.